# GRAPH AS POINT SET

## ABSTRACT

Graphs, fundamental data structures with diverse real-world applications, are composed of *interconnected* nodes. Current Graph Neural Networks (GNNs) have primarily focused on encoding these intricate connections. They employ a message passing framework or complex neural network architectures to handle adjacency matrices, resulting in numerous intricate designs. In contrast, this paper proposes a **paradigm-shifting** approach by introducing a novel graph-to-set conversion method. This innovative technique bijectively transforms interconnected nodes into independent points, suitable for processing by a set encoder, such as the Transformer model. We achieve graph representation learning through the lens of set learning, eliminating the need for intricate positional encodings used in previous graph Transformers. Theoretically, our method outperforms existing models in terms of both short-range and long-range expressivity. Extensive experimental validation further confirms our model's outstanding real-world performance.

## 1 INTRODUCTION

Graphs, composed of interconnected nodes and edges, have wide-ranging applications and have been extensively investigated. In graph machine learning, a central focus is effectively leveraging the adjacency relationships between nodes. Many distinct architectures have arisen for graph tasks, exhibiting significant divergence in their approaches to utilizing the adjacency information.

Two primary paradigms have evolved for encoding adjacency information. The first paradigm involves message passing between nodes via edges. Notable methods in this category include Message Passing Neural Network (MPNN) (Gilmer et al., 2017), a foundational framework for Graph Neural Networks (GNNs) like Graph Convolutional Networks (Kipf & Welling, 2017), Graph Isomorphism Networks (Xu et al., 2019a), and GraphSAGE (Hamilton et al., 2017). These models aggregate messages from neighboring nodes to update the central node's representation. Additionally, subgraph-based GNNs (Zhang & Li, 2021; Huang et al., 2023; Bevilacqua et al., 2022; Qian et al., 2022; Frasca et al., 2022; Zhao et al., 2022; Zhang et al., 2023a) select subgraphs from the whole graph and run MPNN within each subgraph. Simultaneously, Graph Transformers (GTs) integrate adjacency information into the attention matrix (Mialon et al., 2021; Kreuzer et al., 2021; Wu et al., 2021; Dwivedi & Bresson, 2020; Ying et al., 2021; Shirzad et al., 2023), with some recent variants incorporating message-passing layers directly into their architectures (Rampásek et al., 2022; Kim et al., 2021). All these models rely on adjacency relationships to facilitate information exchange among nodes. The second paradigm designs permutation-equivariant neural networks that directly take the adjacency matrix as input. Techniques in this category include high-order Weisfeiler-Leman tests (Maron et al., 2019a), invariant graph neural networks (Maron et al., 2019b), and relational pooling (Chen et al., 2020). Additionally, various studies have explored manual feature extraction from the adjacency matrix, including methods like random walk structural encoding (Dwivedi et al., 2022a), Laplacian matrix eigenvectors (Wang et al., 2022; Lim et al., 2023), and shortest path distances (Li et al., 2020). However, it's important to note that these approaches typically serve as data augmentation steps for other models, rather than constituting an independent paradigm.

This paper deviates from the prevailing paradigms of graph representation learning by introducing an innovative graph-to-set approach as shown in Figure 1: converting interconnected nodes into a set of independent points, subsequently encoded through a set encoder like Transformer. This graph-to-set conversion constitutes a primary contribution of this work. Leveraging our symmetric rank decomposition, we break down the augmented adjacency matrix $A + D$ into $QQ^T$, wherein $Q$ is constituted by column-full-rank rows—each denoting a node coordinate. This representation enables us to

express the presence of edges as inner products of coordinate vectors ($Q_i$ and $Q_j$). Consequently, interlinked nodes can be transformed into independent points and supplementary coordinates without information loss. We theoretically show that two graphs are isomorphic iff the two converted point sets are equal up to an orthogonal transformation. This equivalence empowers us to encode the set with coordinates in an orthogonal-transformation-equivariant manner, in analogy to E(3)-equivariant models designed for 3D geometric deep learning.

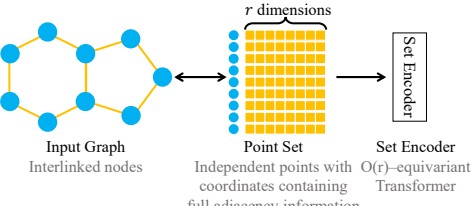

Figure 1: A new paradigm for graph learning. Converting the input graph to a point set first and encoding it with a set encoder. $O(r)$ denotes the set of $r$-dimension orthogonal transformations.

Our work's second contribution is to introduce an orthogonal-transformation-equivariant Transformer to encode the point set. This architecture provably surpasses existing models in both long-range and short-range expressivity. Extensive experiments verify these claims across synthetic datasets, small molecule datasets, and long-range graph benchmarks. Specifically, we outperform all baselines on QM9 (Wu et al., 2017), ZINC (Gómez-Bombarelli et al., 2016), and Long Range Graph Benchmark (Dwivedi et al., 2022b).

In summary, this paper introduces a new paradigm of graph representation learning by converting interconnected graphs into sets of independent points and subsequently encoding them via an orthogonal-transformation-equivariant Transformer. This novel approach outperforms existing methods in both long- and short-range tasks, as validated by comprehensive experiments.

## 2  PRELIMINARY

Given a matrix $Z \in \mathbb{R}^{a \times b}$, we define $Z_i \in \mathbb{R}^b$ as the $i$-th row of $Z$ in the form of a column vector, and $Z_{ij} \in \mathbb{R}$ as the element of $Z$ in the $(i, j)$ position. Considering a vector $\Lambda \in \mathbb{R}^a$, we define $\text{diag}(\Lambda) \in \mathbb{R}^{a \times a}$ as the diagonal matrix with $\Lambda$ as its diagonal elements. Moreover, for a matrix $S \in \mathbb{R}^{a \times a}$, we define $\text{diagonal}(S) \in \mathbb{R}^a$ as the vector of the diagonal elements of $S$.

Let $\mathcal{G} = (V, E, X)$ denote an *undirected graph.* Here, $V = \{1, 2, 3, ..., n\}$ represents the set of $n$ nodes, $E \subseteq V \times V$ is the set of edges, and $X \in \mathbb{R}^{n \times d}$ is the node feature matrix. The $v$-th row $X_v$ of $X$ is the features of node $v$. The edge set $E$ can also be represented using the adjacency matrix $A \in \mathbb{R}^{n \times n}$, where $A_{uv}$ is 1 if the edge exists (i.e., $(u, v) \in E$) and 0 otherwise. A graph $\mathcal{G}$ can also be represented by the pair $(V, A, X)$ or $(A, X)$. The degree matrix $D$ is a diagonal matrix with node degree (sum of corresponding rows of matrix $A$) as the diagonal elements.

Given a permutation function $\pi : \{1, 2, 3, ..., n\} \rightarrow \{1, 2, 3, ..., n\}$, the permuted graph is $\pi(\mathcal{G}) = (\pi(A), \pi(X))$. Here $\pi(A) \in \mathbb{R}^{n \times n}$, and for all nodes $u$ and $v$ in $V$, $\pi(A)_{\pi(u)\pi(v)} = A_{uv}$. Moreover, $\pi(X) \in \mathbb{R}^{n \times d}$, where $\pi(X)_{\pi(v)} = X_v$. Essentially, the permutation $\pi$ reindex each node $v$ to $\pi(v)$ while preserving the original graph structure and node features. Two graphs are isomorphic iff they can be mapped to each other through a permutation.

**Definition 1.** *Graphs $\mathcal{G}_1 = (A_1, X_1)$ and $\mathcal{G}_2 = (A_2, X_2)$ are isomorphic, denoted as $\mathcal{G}_1 \simeq \mathcal{G}_2$, if there exists a permutation $\pi$ such that $\pi(\mathcal{G}_1) = \mathcal{G}_2$ (equivalent to $\pi(A_1) = A_2$ and $\pi(X_1) = X_2$).*

Isomorphic graphs can be transformed into each other by merely reindexing their nodes. In tasks involving whole graph classification, models should assign the same prediction to isomorphic graphs.

**Symmetric Rank Decomposition (SRD)**    Decomposing an arbitrary matrix into two full-rank matrices is a well-known technique (Puntanen et al., 2011). We further extend this technique to demonstrate that a positive semi-definite matrix can be decomposed into a full-rank matrix.

**Definition 2.** *(Symmetric Rank Decomposition, SRD) Given a (symmetric) positive semi-definite matrix $L \in \mathbb{R}^{n \times n}$ of rank $r$, the SRD of $L$ is given by $L = QQ^T$, where $Q \in \mathbb{R}^{n \times r}$.*

As $L = QQ^T$, it follows that $rank(Q) = rank(L) = r$, which implies that $Q$ must be full column rank. Moreover, two SRDs of the same matrix are equivalent up to an orthogonal transformation. Let $O(r)$ denote the set of orthogonal matrices in $\mathbb{R}^{r \times r}$.

**Proposition 1.** *Matrices $Q_1$ and $Q_2$ in $\mathbb{R}^{n \times r}$ represent the SRD of the same matrix if and only if there exists an orthogonal matrix $R \in O(r)$ such that $Q_1 = Q_2 R$.*

SRD is closely related to eigendecomposition. Let $L = U \text{diag}(\Lambda) U^T$ denote the eigendecomposition of $L$, where $\Lambda \in \mathbb{R}^r$ is the vector of non-zero eigenvalues, and $U \in \mathbb{R}^{n \times r}$ is the matrix whose columns are the corresponding eigenvectors. Setting $Q = U \text{diag}(\Lambda^{1/2})$ yields an SRD of $L$, where the superscript denotes element-wise square root operation.

## 3 GRAPH AS POINT SET

In this section, we present our innovative method for converting graphs into sets of points. We first demonstrate that the Symmetric Rank Decomposition (SRD) can theoretically achieve this transformation: two graphs are isomorphic if and only if the sets of coordinates generated by SRD are equal up to orthogonal transformations. Additionally, we parameterize the SRD for better real-world performance. All proofs for the theorems in this section can be found in Appendix A.

### 3.1 SYMMETRIC RANK DECOMPOSITION FOR COORDINATES

A natural approach to breaking down the interconnections between nodes is to decompose the adjacency matrix. While previous methods utilize the outputs of eigendecomposition as supplementary node features, these features are not unique. Consequently, models naively utilizing such features fail to provide consistent predictions for isomorphic or even identical graphs, ultimately leading to poor generalization. To address this, we show that Symmetric Rank Decomposition (SRD) can convert graph-level tasks into set-level tasks with perfect alignment.

Since SRD only applies to positive semi-definite matrices, we use the augmented adjacency matrix $D + A$, which is always positive semi-definite (see Appendix A.2 for the proof).

**Theorem 1.** *Given two graphs $\mathcal{G} = (V, A, X)$ and $\mathcal{G}' = (V', A', X')$ with respective degree matrices $D$ and $D'$, the two graphs are isomorphic ($\mathcal{G} \simeq \mathcal{G}'$) if and only if $\exists R \in O(r), \{\!\{(X_v, RQ_v) | \forall v \in V\}\!\} = \{\!\{(X_v', Q_v') | v \in V'\}\!\}$, where $r$ denotes the rank of matrix $A$, and $Q$ and $Q'$ are the symmetric rank decompositions of $D + A$ and $D' + A'$ respectively.*

In this theorem, the graph $\mathcal{G} = (V, A, X)$ is converted to a set of points $\{(X_v, Q_v) | v \in V\}$, where $X_v$, the $v$-th row of node feature matrix $X$, is the original node feature of $v$, and $Q_v$, the $v$-th row of SRD of $D + A$, is the $r$-dimensional coordinate of node $v$. Consequently, two graphs are isomorphic if their sets of points are equal up to an orthogonal transformation. Intuitively, we can imagine that we map the graph into an $r$-dimensional space, where each node is associated with a coordinate, and the inner product between two coordinates define the existence of an edge. Nevertheless, this mapping is not unique, since we can arbitrarily rotate the coordinates through an $O(r)$ matrix without changing the inner products. This conversion can be roughly understood as an **inverse process** of 3D-point-set-to-graph transformation in geometric deep learning, where Euclidean distances between points are used to define edges between nodes. Many molecular graphs are constructed this way. Note that here the analogy is only for intuitive understanding and not a precise equivalence. Our primary focus is graph learning rather than its connection with 3D geometric deep learning.

Leveraging Theorem 1, we can transform a graph into a set and use a set encoder, such as Transformer, to encode the graph. Notably, our method generates consistent representations for isomorphic graphs if the encoder remains invariant to orthogonal transformations. Furthermore, the expressivity of the entire method depends on the set encoder's expressivity to distinguish non-equal sets. The more powerful the set encoder is, the more expressive the whole method is on graph tasks.

### 3.2 GENERALIZED COORDINATES

The Symmetric Rank Decomposition (SRD) offers a theoretical pathway to convert graph problems into set problems perfectly. In this section, we further parameterize SRD for better performance in practice. As shown in Section 2, SRD can be implemented using eigendecomposition, where $Q = U \text{diag}(\Lambda^{1/2})$, with $\Lambda \in \mathbb{R}^r$ is the vector of non-zero eigenvalues, and $U \in \mathbb{R}^{n \times r}$ are the corresponding eigenvectors. To generalize it, we can replace the element-wise square root with a permutation-equivariant function $f : \mathbb{R}^r \to \mathbb{R}^r$. This modification also removes the constraint that

all eigenvalues must be non-negative. Now, various symmetric matrices that contain adjacency information, like the adjacency matrix $A$ and the normalized adjacency matrix $\hat{A} = D^{-1/2}AD^{-1/2}$., can be used to produce coordinates. Let $\tilde{Q}(Z, f) = Uf(\Lambda) \in \mathbb{R}^{n \times r}$ denote the **generalized coordinates**, where $U, \Lambda$ are produced by the eigendecomposition of the matrix $Z$. Given a fixed $f$, the two sets with generalized coordinates are still equal up to orthogonal transformations if two graphs are isomorphic. Moreover, the reverse is true when $f$ is expressive enough.

**Theorem 2.** *Given two graphs $\mathcal{G} = (V, A, X)$ and $\mathcal{G} = (V', A', X')$, and an injective permutation-equivariant function $\mathcal{Z}$ mapping adjacency matrix to a symmetric matrix: (1) For all permutation-equivariant function $f$, if $\mathcal{G} \simeq \mathcal{G}'$, then the two sets of generalized coordinates are equal up to an orthogonal transformation, i.e., $\exists R \in O(r), \{\!\!\{ X_v, R\tilde{Q}(\mathcal{Z}(A), f)_v | v \in V \}\!\!\} = \{\!\!\{ X'_v, \tilde{Q}(\mathcal{Z}(A'), f)_v | v \in V' \}\!\!\}$, where $r$ is the rank of $A$, $\tilde{Q}, \tilde{Q}'$ are the generalized coordinates of $A$ and $A'$ respectively. (2) There exists a continuous permutation-equivariant function $f : \mathbb{R}^r \to \mathbb{R}^{r \times 2}$, such that $\mathcal{G} \simeq \mathcal{G}'$ if $\exists R \in O(r), \forall i = 1, 2, ..., d, \{\!\!\{ (X_v, R\tilde{Q}(\mathcal{Z}(A), f_1)_v, R\tilde{Q}(\mathcal{Z}(A), f_2)_v) | v \in V \}\!\!\} = \{\!\!\{ (X'_v, \tilde{Q}(\mathcal{Z}(A'), f_1)_v, \tilde{Q}(\mathcal{Z}(A'), f_2)_v) | v \in V' \}\!\!\}$, where $f_1 : \mathbb{R}^r \to \mathbb{R}^r$ and $f_2 : \mathbb{R}^r \to \mathbb{R}^r$ are two output channels of $f$.*

Therefore, we can use arbitrary permutation-equivariant functions to transform the eigenvalues without worrying about producing different predictions for isomorphic graphs. Moreover, with an expressive enough eigenvalue function, the graph-level tasks can be converted to set problems perfectly. In implementation, we use DeepSet (Segol & Lipman, 2020) due to its universal expressivity for permutation-equivariant functions. The output coordinates will have multiple channels, each channel corresponds to an output dimension of the eigenvalue function. The detailed Architecture is shown in Figure 3 in Appendix G. In summary, we use SRD and its parameterized generalization to decompose the adjacency matrix or its variants into coordinates. Thus, we transform a graph into a point set in which each point corresponds to a node and has the original node feature and the coordinates as its feature.

# 4 POINT SET TRANSFORMER

As shown in Figure 1, our method consists of two steps: converting graph to a set of independent points and encoding the set. Section 3.1 has illustrated how to transform the graph to a set bijectively. To encode the transformed point set, this section introduces a novel transformer architecture, which maintains invariance to orthogonal transformations and achieves outstanding expressivity.

The overall architecture is depicted in Figure 4 in Appendix G. Our transformer operates with nodes carrying two types of representations: scalars, which remain invariant to coordinate orthogonal transformations, and vectors, which adapt equivariantly to coordinate changes. Scalars are initialized using input node features, while vectors draw from the coordinate values. For a point $i$, its scalar representation is denoted by $s_i \in \mathbb{R}^d$, and its vector representation is denoted by $v_i \in \mathbb{R}^{d \times r}$, where $d$ is the hidden dimension, and $r$ is the rank of the decomposed matrix of the graph (Laplacian matrix by default in this model). $s_i$ is initialized with the node feature $X_i$, and $v_i$ is initialized with the parameterized coordinates containing graph structural information, as detailed in Section 3.2.

Our point set transformer (PST) is composed of several transformer layers. Each layer involves two key components analogous to ordinary transformer structures:

**Scalar-Vector Mixer.** This component, akin to the feed-forward networks in transformers, individually transforms point representations. To enable the information exchange between vector and scalar features, we design a mixer architecture as follows.

$$s'_i \leftarrow \text{MLP}_1(s_i \| \text{diagonal}(W_1 v_i v_i^T W_2^T)), \quad v'_i \leftarrow W_3 \text{diag}(\text{MLP}_2(s_i))v_i + W_4 v_i \quad (1)$$

Here, $W_1, W_2, W_3$, and $W_4 \in \mathbb{R}^{d \times d}$ are learnable matrices for mixing different channels of vector features. Additionally, $\text{MLP}_1 : \mathbb{R}^{2d \to d}$ and $\text{MLP}_2 : \mathbb{R}^{d \to d}$ represent two multi-layer perceptrons transforming scalar representations. The operation $\text{diagonal}(W_1 v_i v_i^T W_2)$ takes the diagonal elements of a matrix, which translates vectors to scalars, while $\text{diag}(\text{MLP}_2(s_i))v_i$ transforms scalar features into vectors. As $v_i RR^T v_i^T = v_i v_i^T, \forall R \in O(r)$, the scalar update is invariant to orthogonal transformations of the coordinates. Similarly, the vector update is equivariant to $O(r)$.

**Attention Layer.** Similar to ordinary attention layer, this component computes inner products between representations of point pairs to linearly combine node representations.

$$\text{Atten}_{ij} = \text{MLP}(K_{ij}), \quad K_{ij} = (W_q^s s_i \odot W_k^s s_j) \| \text{diagonal}(W_q^v v_i v_j^T W_k^v) \tag{2}$$

Here, $W_q^s$ and $W_q^v$ denote the linear transformations for scalars and vectors queries, respectively, while $W_k^s$ and $W_k^v$ are for keys. The equation computes the inner products of queries and keys, similar to standard attention mechanisms. It is easy to see $\text{Atten}_{ij}$ is also invariant to $O(r)$.

Following the calculation of the attention matrix, we multiply it with point representations:

$$s_i \leftarrow \sum_j \text{Atten}_{ij} s_j', \quad v_i \leftarrow \sum_j \text{Atten}_{ij} v_j' \tag{3}$$

Each transformer layer is of time complexity $O(n^2 r)$ and space complexity $O(n^2 + nr)$.

**Pooling.** After several layers, we pool all points' scalar representations as the set representation $s$.

$$s \leftarrow \text{Pool}(\{s_i | i \in V\}), \tag{4}$$

where Pool is pooling function like sum, mean, and max.

## 5 EXPRESSIVITY

Though our primary focus is a new paradigm for graph representation learning, we also delve into the theoretical expressivity of our methods in this section. Our generalized coordinates and the PST architecture exhibit strong long-range expressivity, allowing for efficient computation of distance metrics between nodes, as well as short-range expressivity, enabling the counting of paths and cycles rooted at each node. Therefore, our model is more expressive than many existing models, including GIN (equivalent to the 1-WL test) (Xu et al., 2019b), PPGN (equivalent to the 2-FWL test, more expressive in some cases) (Maron et al., 2019a), GPS (Rampásek et al., 2022), and Graphormer (Ying et al., 2021) (two representative graph transformers). More details are in Appendix B.

### 5.1 LONG RANGE EXPRESSIVITY

This section demonstrates that the inner product of generalized coordinates exhibits strong long-range expressivity. PST inherits this strong expressivity, since it utilizes these inner products.

When assessing a model's capacity to capture long-range interactions (LRI), an essential metric is its ability to compute shortest path distances (spd) between nodes. Since formally characterizing LRI can be challenging, we focus on analyzing models' performance concerning this specific metric. We observe that existing models vary significantly in this metric (e.g., MPNNs even cannot express spd). Moreover, we find an intuitive way to explain these differences: Shortest path distances between nodes are calculated using the expression $spd(i, j, A) = \arg\min_k\{k | A_{ij}^k > 0\}$, and the ability to compute $A^K$, the $K$-th power of the adjacency matrix $A$, can serve as an intuitive indicator of a model's capacity for computing spd, yet different models need different # layers to compute $A^K$.

**Generalized Coordinates.** Generalized coordinates can capture arbitrarily large shortest path distances through their inner products in one step. To illustrate it, we decompose the adjacency matrix as $A = U\Lambda U^T$, and employ coordinates as $U$ and $U\text{diag}(\Lambda^K)$. Their inner products are as follows:

$$\overbrace{U\text{diag}(\Lambda^K)U^T \to A^K}^{1 \text{ step}} \tag{5}$$

**Theorem 3.** *For all undirected graphs $\mathcal{G} = (A, X)$, there exists permutation-equivariant functions $f_k, k = 0, 1, 2, ..., K$, such that the shortest path distance between node $i, j$ is a function of $\langle \tilde{Q}(A, f_0)_i, \tilde{Q}(A, f_k)_j \rangle$, $k = 0, 1, 2, ...K$, where $\tilde{Q}(A, f)$ is the generalized coordinates defined in Section 3.2, $K$ is the maximum shortest path distance between nodes (the diameter of graph).*

**2-FWL.** A powerful graph isomorphic test, 2-Folklore-Weisfeiler-Leman Test (2-FWL) (Maron et al., 2019a), produces node pair representations in a matrix $X \in \mathbb{R}^{n \times n}$. $X$ is initialized with $A$. Each layer updates $X$ with $XX$. So intuitively, computing $A^K$ takes $\lceil \log_2 K \rceil$ layers.

$$\overbrace{A \to A^2 = (A)(A) \to A^4 = (A^2)(A^2) \to A^8 = (A^4)(A^4) \to ... \to A^K = (A^{K/2} A^{K/2})}^{\lceil \log_2 K \rceil \text{ layers}} \tag{6}$$

Formally, we state the following theorem:

**Theorem 4.** *For all $K \in \mathbb{N}^+$, for all undirected graphs $\mathcal{G} = (A, X)$ and $\mathcal{G}' = (A', X')$, let $c(\mathcal{G})_{ij}^k$ denote the color of node tuple $(i, j)$ of graph $\mathcal{G}$ at iteration $k$. For all node tuples $(i, j)$ and $(i', j')$, if $spd(i, j, A) < spd(i', j', A') \leq 2^K$, then $c(\mathcal{G})_{ij}^K \neq c(\mathcal{G}')_{i'j'}^K$. Moreover, for all $L > 2^K$, there exists $i, j, i', j'$, such that $spd(i, j, A) > spd(i', j', A') = L$ while $c(\mathcal{G})_{ij}^K = c(\mathcal{G}')_{i'j'}^K$.*

In other words, using $K$ iterations of 2-FWL allows us to distinguish between pairs of nodes with different spds, as long as that distance is at most $2^K$. Moreover, $K$-iteration 2-FWL cannot differentiate all tuples with arbitrary spd $> 2^K$ from other tuples with different spds, which indicates that $K$-iteration 2-FWL is effective in counting shortest path distances up to a maximum of $2^K$.

**MPNNs.** Intuitively, each MPNN layer uses $AX$ to update node representations $X$. However, this operation in general cannot compute $A^K$ unless the initial node feature $X = I$.

$$\overbrace{X \rightarrow AX \rightarrow A^2X = A(AX) \rightarrow A^3X = A(A^2X) \rightarrow ... \rightarrow A^KX = A(A^{K-1})X}^{K \text{ layers}} \quad (7)$$

Formally, we state the following theorem which indicates that MPNNs cannot compute spds:

**Theorem 5.** *A graph pair exists that MPNN cannot differentiate, but their set of all-pair shortest path distances are different.*

If MPNNs can compute spds between all pairs of nodes, then from the set of spds they should be able to distinguish this graph pair. However, we show no MPNNs can distinguish the pair, thus proving that MPNNs are unable to compute spds.

It is known that Graph Transformers (GTs) possess a strong capacity for capturing long-range interactions (Dwivedi et al., 2022b), given their ability to aggregate information from the entire graph to update each node's representation. However, note that aggregating information from the entire graph is not equivalent to capturing the distance between nodes, and some Graph Transformers fail to compute the shortest path distance between nodes. Details are in Appendix C. Note that this slightly counter-intuitive results is because we take a new perspective to study long range interaction rather than showing GTs are weak in long range capacity.

Besides shortest path distances, our generalized coordinates are parameterized, enabling the unification of various structure encodings, encompassing various distance metrics between nodes, including random walk (Li et al., 2020; Dwivedi et al., 2023; Rampásek et al., 2022), heat kernel (Mialon et al., 2021), resistance distance (Zhang & Li, 2021; Zhang et al., 2023b). For further insights and details about the versatility of our generalized coordinates, please refer to Table 5 in Appendix D.

### 5.2 Short Range Expressivity

This section shows PST's expressivity in representative short-range tasks: path and cycle counting.

**Theorem 6.** *With sufficiently large hidden dimensions, a one-layer Point Set Transformer can count paths of length $1$ and $2$, a two-layer model can count paths of length $3$ and $4$, and a four-layer model can further count paths of length $5$ and $6$. Here, "count" means that the $(i, j)$ element of the attention matrix in the last layer can express the number of paths between nodes $i$ and $j$.*

Therefore, with enough layers, our PST models can count the number of paths of length $\leq 6$ between nodes. Furthermore, our PST can also count cycles.

**Theorem 7.** *With sufficiently large hidden dimensions, a one-layer Point Set Transformer can count cycles of length $3$, a three-layer model can count cycles of length $4$ and $5$, and a five-layer model can further count cycles of length $6$ and $7$. Here, "count" means the representation of node $i$ in the last layer can express the number of cycles involving node $i$.*

Therefore, with enough layers, our PST models can count the number of cycles of length $\leq 7$ between nodes. Given that even 2-FWL is restricted to counting cycles up to length 7, the cycle counting power of our Point Set Transformer is at least on par with 2-FWL.

## 6 Related Work

**Graph Neural Network with Eigen-Decomposition.** Our approach, the Point Set Transformer, leverages coordinates derived from the symmetric rank decomposition (SRD) of adjacency or re-

lated matrices as input. In contrast, prior studies primarily focus on eigendecomposition (EVD) techniques. Although two approaches share similarities, our SRD offers a distinct advantage by transforming the graph isomorphism problem into a set problem **bijectively**—an achievement not possible through EVD. This fundamental divergence in theoretical capabilities has significant implications for model design. Early efforts by Dwivedi et al. (2023) introduce eigenvectors to augment input node features in MPNNs (Gilmer et al., 2017). Subsequent works, such as Graph Transformers (Dwivedi & Bresson, 2020; Kreuzer et al., 2021), incorporate eigenvectors as positional node encodings. However, due to the non-uniqueness of eigenvectors, these models generate disparate predictions even for isomorphic graphs, leading to poor generalization. Lim et al. (2023) propose sign-and-basis-invariant networks to address this, but these solutions only apply when eigenvalue multiplicity remains constant, limiting their applicability in graph tasks. Wang et al. (2022) and Bo et al. (2023) use eigenvectors to produce edge features. They solves non-uniqueness, but these methods are still essentially MPNNs. In stark contrast, we convert graph-level tasks into set-level tasks with perfect alignment and thus propose a new paradigm of graph representation learning: use a orthogonal-transformation-equivariant set encoder to handle the transformed point set.

**Equivariant Point Cloud and 3-D Molecule Neural Networks.** Equivariant point cloud and 3-D molecule tasks share resemblances: both involve unordered sets of 3-D coordinate points as input and require models to produce predictions invariant/equivariant to orthogonal transformations and translations of coordinates. Several works (Chen et al., 2021; Winkels & Cohen, 2018; Cohen et al., 2018; Gasteiger et al., 2021) introduce specialized equivariant convolution operators to preserve prediction symmetry, yet are later surpassed by models that learn both invariant and equivariant representations for each point, transmitting these representations between nodes. Notably, certain models (Satorras et al., 2021; Schütt et al., 2021; Deng et al., 2021; Wang & Zhang, 2022) directly utilize vectors mirroring input coordinate changes as equivariant features, while others (Thomas et al., 2018; Batzner et al., 2022; Fuchs et al., 2020; Hutchinson et al., 2021; Worrall et al., 2017; Weiler et al., 2018) incorporate high-order irreducible representations of the orthogonal group, achieving proven universal expressivity (Dym & Maron, 2021). Our Point Set Transformer (PST) similarly learns both invariant and equivariant point representations. However, due to the specific conversion of point sets from graphs, PST's architecture varies from existing models. While translation invariance characterizes point clouds and molecules, graph properties are sensitive to coordinate translations. Hence, we adopt inner products of coordinates. Additionally, these prior works center on 3D point spaces, whereas our coordinates exist in high-dimensional space, rendering existing models and theoretical expressivity results based on high-order irreducible representations incompatible with our framework.

## 7 EXPERIMENTS

In our experiments, we validate the effectiveness of our model across three key dimensions: substructure counting tasks (Huang et al., 2023) to demonstrate theoretical short-range expressiveness, real-world small graph properties prediction tasks (Wu et al., 2017; Gómez-Bombarelli et al., 2016; Hu et al., 2020) to assess experimental expressiveness, and Long-Range Graph Benchmarks (Dwivedi et al., 2022b) to evaluate its ability to capture long-range interactions. Additionally, our generalized coordinates is validated to be effective on both long and short range tasks in ablation study (see Section H). Our PST also have a similar scalability to our baselines (see Section I). For all dataset, our PST uses less or comparable number of parameter compared with baselines. Details of datasets and experiment settings could be found in Appendix E and Appendix F.

### 7.1 GRAPH SUBSTRUCTURE COUNTING

As highlighted in previous research (Chen et al., 2020), the ability to count substructures is a critical metric for assessing expressivity. Therefore, we evaluate our model's capabilities in substructure counting tasks on synthetic graphs following Huang et al. (2023). The considered substructures encompass paths of lengths 2 to 6, cycles of lengths 3 to 7, and other substructures including tailed triangles (TT), chordal cycles (CC), and triangle-rectangle (TR). Our task involves predicting the number of paths originating from each node and the cycles and other substructures in which each node participates. For comparison, we evaluate our Point Set Transformer (PST) alongside other

Table 1: We evaluate the normalized Mean Absolute Error (MAE) (↓) of substructure counting tasks on a synthetic dataset. Similar to the approach in Huang et al. (2023), models successfully count the corresponding structure if the test loss is below 10 units (yellow cell in the table), measured using a scale of $10^{-3}$. TT means Tailed Triangle. CC means Chordal Cycle, TR means Triangle-Rectangle.

| Method | 2-Path | 3-Path | 4-Path | 5-path | 6-path | 3-Cycle | 4-Cycle | 5-Cycle | 6-Cycle | 7-cycle | TT | CC | TR |
|---|---|---|---|---|---|---|---|---|---|---|---|---|---|
| MPNN | 1.0 | 67.3 | 159.2 | 235.3 | 321.5 | 351.5 | 274.2 | 208.8 | 155.5 | 169.8 | 363.1 | 311.4 | 297.9 |
| IDGNN | 1.9 | 1.8 | 27.3 | 68.6 | 78.3 | 0.6 | 2.2 | 49 | 49.5 | 49.9 | 105.3 | 45.4 | 62.8 |
| NGNN | 1.5 | 2.1 | 24.4 | 75.4 | 82.6 | 0.3 | 1.3 | 40.2 | 43.9 | 52.2 | 104.4 | 39.2 | 72.9 |
| GNNAK | 4.5 | 40.7 | 7.5 | 47.9 | 48.8 | 0.4 | 4.1 | 13.3 | 23.8 | 79.8 | 4.3 | 11.2 | 131.1 |
| $I^2$-GNN | 1.5 | 2.6 | 4.1 | 54.4 | 63.8 | 0.3 | 1.6 | 2.8 | 8.2 | 39.9 | 1.1 | 1.0 | 1.3 |
| PPGN | 0.3 | 1.7 | 4.1 | 15.1 | 21.7 | 0.3 | 0.9 | 3.6 | 7.1 | 27.1 | 2.6 | 1.5 | 14.4 |
| PST | $0.7_{\pm0.1}$ | $1.1_{\pm0.1}$ | $1.5_{\pm0.1}$ | $2.2_{\pm0.1}$ | $3.3_{\pm0.3}$ | $0.8_{\pm0.1}$ | $1.9_{\pm0.2}$ | $3.1_{\pm0.3}$ | $4.9_{\pm0.3}$ | $8.6_{\pm0.5}$ | $3.0_{\pm0.1}$ | $4.0_{\pm0.7}$ | $9.2_{\pm0.9}$ |

Table 2: Test Mean Absolute Error (MAE) for property prediction on the QM9 dataset. LRP represents Deep LRP (Chen et al., 2020). DF represents 2-DRFWL(2) GNN (Zhou et al., 2023). 1GNN and 123 correspond to 1-GNN and 1-2-3-GNN (Morris et al., 2019), respectively. * denotes models with 3D coordinates or features as input.

| Target | Unit | LRP | NGNN | $I^2$GNN | DF | PST | 1GNN* | DTNN* | 123* | PPGN* | PST* |
|---|---|---|---|---|---|---|---|---|---|---|---|
| $\mu$ | $10^{-1}$D | 3.64 | 4.28 | 4.28 | 3.46 | $\mathbf{3.19}_{\pm0.04}$ | 4.93 | 2.44 | 4.76 | 2.31 | $\mathbf{0.23}_{\pm0.01}$ |
| $\alpha$ | $10^{-1}a_0^3$ | 2.98 | 2.90 | 2.30 | 2.22 | $\mathbf{1.89}_{\pm0.04}$ | 7.80 | 9.50 | 2.70 | 3.82 | $\mathbf{0.78}_{\pm0.05}$ |
| $\varepsilon_{\text{homo}}$ | $10^{-2}$meV | 6.91 | 7.21 | 7.10 | 6.15 | $\mathbf{5.98}_{\pm0.09}$ | 8.73 | 0.1056 | 9.17 | 7.51 | $\mathbf{2.59}_{\pm0.08}$ |
| $\varepsilon_{\text{lumo}}$ | $10^{-2}$meV | 7.54 | 8.08 | 7.27 | 6.12 | $\mathbf{5.84}_{\pm0.08}$ | 9.66 | 0.1393 | 9.55 | 7.81 | $\mathbf{2.20}_{\pm0.07}$ |
| $\Delta\varepsilon$ | $10^{-2}$meV | 9.61 | 10.34 | 10.34 | 8.82 | $\mathbf{8.46}_{\pm0.07}$ | 13.33 | 30.48 | 13.06 | 11.05 | $\mathbf{4.47}_{\pm0.09}$ |
| $R^2$ | $a_0^2$ | 19.30 | 20.50 | 18.64 | 15.04 | $\mathbf{13.08}_{\pm0.16}$ | 34.10 | 17.00 | 22.90 | 16.07 | $\mathbf{0.93}_{\pm0.03}$ |
| ZPVE | $10^{-2}$meV | 1.50 | 0.54 | 0.38 | 0.46 | $\mathbf{0.39}_{\pm0.01}$ | 3.37 | 4.68 | 0.52 | 17.42 | $\mathbf{0.26}_{\pm0.01}$ |
| $U_0$ | meV | 11.24 | 8.03 | 5.74 | 4.24 | $\mathbf{3.46}_{\pm0.17}$ | 63.13 | 66.12 | **1.16** | 6.37 | $3.33_{\pm0.19}$ |
| $U$ | meV | 11.24 | 9.82 | 5.61 | 4.16 | $\mathbf{3.55}_{\pm0.10}$ | 56.60 | 66.12 | **3.02** | 6.37 | $3.26_{\pm0.05}$ |
| $H$ | meV | 11.24 | 8.30 | 7.32 | 3.95 | $\mathbf{3.49}_{\pm0.20}$ | 60.68 | 66.12 | **1.14** | 6.23 | $3.29_{\pm0.21}$ |
| $G$ | meV | 11.24 | 13.31 | 7.10 | 4.24 | $\mathbf{3.55}_{\pm0.17}$ | 52.79 | 66.12 | **1.28** | 6.48 | $3.25_{\pm0.15}$ |
| $C_v$ | $10^{-2}$cal/mol/K | 12.90 | 17.40 | **7.30** | 9.01 | $7.77_{\pm0.15}$ | 27.00 | 243.00 | 9.44 | 18.40 | $\mathbf{3.63}_{\pm0.13}$ |

expressive GNN models, such as ID-GNNs (You et al., 2021), NGNNs (Zhang & Li, 2021), GN-NAK+(Zhao et al., 2022), $I^2$-GNN(Huang et al., 2023), and PPGN (Maron et al., 2019a).

The results are shown in Table 1. Following the criteria established by Huang et al. (2023), we consider a model able to count a specific substructure if its normalized test Mean Absolute Error (MAE) is below $10^{-2}$ (10 units in the table). Remarkably, our PST demonstrates can count all substructures listed in the table, whereas the second-best model, $I^2$-GNN, can only count 10 out of the 13 substructures in total. The low loss on counting cycles and paths aligns well with our theoretical results (Theorem 6 and Theorem 7).

## 7.2 MOLECULAR PROPERTIES PREDICTION

To validate the real-world effectiveness of our Point Set Transformer (PST), we conduct experiments on four well-known molecular graph datasets: QM9 (Wu et al., 2017), ZINC, ZINC-full (Gómez-Bombarelli et al., 2016), and ogbg-molhiv (Hu et al., 2020). For the QM9 dataset, we use various expressive GNNs as baseline models. Some of these models incorporate the Euclidean distance between atoms in 3D space, including 1-GNN, 1-2-3-GNN (Morris et al., 2019), DTNN (Wu et al., 2017), and PPGN (Maron et al., 2019a). Others focus solely on learning the graph structure without considering 3D atom coordinates, such as Deep LRP (Chen et al., 2020), NGNN (Zhang & Li, 2021), $I^2$-GNN (Huang et al., 2023), and 2-DRFWL(2) GNN (Zhou et al., 2023). To ensure a fair comparison, we introduce two versions of our model: PST without Euclidean distance (PST) and PST with Euclidean distance (PST*). Baseline results are obtained from (Zhou et al., 2023), and the comprehensive findings are presented in Table 2. Notably, PST outperforms all baseline models without Euclidean distance on 11 out of 12 targets and achieves an average 11% reduction in loss compared to the strongest baseline, 2-DRFWL(2) GNN. Meanwhile, PST* outperforms all Euclidean distance-based baselines on 8 out of 12 targets and achieves an average 4% reduction in loss compared to the strongest baseline, 1-2-3-GNN. Both models also rank second in performance for the remaining targets.

For the ZINC, ZINC-full, and ogbg-molhiv datasets, we have conducted an evaluation of our Point Set Transformer (PST) in comparison to a range of expressive GNNs and graph transformers. This set of models includes GIN (Xu et al., 2019b), GNNAK+(Wang & Zhang, 2022), ESAN(Bevilacqua et al., 2022), SUN (Frasca et al., 2022), SSWL (Zhang et al., 2023a), 2-DRFWL(2) GNN (Zhou et al., 2023), CIN (Bodnar et al., 2021), NGNN (Zhang & Li, 2021), Graphormer (Ying et al., 2021), GPS (Rampásek et al., 2022), Graph MLP-Mixer (He et al., 2023), SAN (Kreuzer et al., 2021), Specformer (Bo et al., 2023), SignNet (Lim et al., 2023), and Grit (Ma et al., 2023). Performance results for the expressive GNNs are sourced from (Zhou et al., 2023), while those for the Graph Transformers are extracted from (He et al., 2023; Ma et al., 2023; Lim et al., 2023). The comprehensive results are presented in Table 3.

Table 3: Results on small molecule datasets with 500k parameter budget.

|  | zinc MAE↓ | zinc-full MAE↓ | molhiv AUC↑ |
|---|---|---|---|
| GIN | $0.163_{\pm0.004}$ | $0.088_{\pm0.002}$ | $77.07_{\pm1.49}$ |
| GNN-AK+ | $0.080_{\pm0.001}$ | – | $79.61_{\pm1.19}$ |
| ESAN | $0.102_{\pm0.003}$ | $0.029_{\pm0.003}$ | $78.25_{\pm0.98}$ |
| SUN | $0.083_{\pm0.003}$ | $0.024_{\pm0.003}$ | $80.03_{\pm0.55}$ |
| SSWL | $0.083_{\pm0.003}$ | $0.022_{\pm0.002}$ | $79.58_{\pm0.35}$ |
| DRFWL | $0.077_{\pm0.002}$ | $0.025_{\pm0.003}$ | $78.18_{\pm2.19}$ |
| CIN | $0.079_{\pm0.006}$ | $0.022_{\pm0.002}$ | $\mathbf{80.94_{\pm0.57}}$ |
| NGNN | $0.111_{\pm0.003}$ | $0.029_{\pm0.001}$ | $78.34_{\pm1.86}$ |
| Graphormer | $0.122_{\pm0.006}$ | $0.052_{\pm0.005}$ | $80.51_{\pm0.53}$ |
| GPS | $0.070_{\pm0.004}$ | - | $78.80_{\pm1.01}$ |
| GMLP-Mixer | $0.077_{\pm0.003}$ | - | $79.97_{\pm1.02}$ |
| SAN | $0.139_{\pm0.006}$ | - | $77.75_{\pm0.61}$ |
| Specformer | $0.066_{\pm0.003}$ | - | $78.89_{\pm1.24}$ |
| SignNet | $0.084_{\pm0.006}$ | $0.024_{\pm0.003}$ | - |
| Grit | $0.059_{\pm0.002}$ | $0.024_{\pm0.003}$ | - |
| PST | $0.063_{\pm0.003}$ | $\mathbf{0.018_{\pm0.001}}$ | $80.32_{\pm0.71}$ |

Notably, our PST consistently outperforms all baseline models on the ZINC and ZINC-full datasets, achieving reductions in loss of 5% and 18%, respectively. On the ogbg-molhiv dataset, our PST also delivers competitive results, with only CIN and Graphormer surpassing it. Overall, the Point Set Transformer demonstrates exceptional performance across these four diverse datasets.

## 7.3 LONG RANGE GRAPH BENCHMARK

To evaluate the long-range capacity of our Point Set Transformer (PST), we conducted experiments using the Long Range Graph Benchmark (Dwivedi et al., 2022b). Following He et al. (2023), we compared our model to a range of baseline models, including GCN (Kipf & Welling, 2017), GINE (Xu et al., 2019a), GatedGCN (Bresson & Laurent, 2017), SAN (Kreuzer et al., 2021), Graphormer (Ying et al., 2021), GMLP-Mixer, Graph ViT (He et al., 2023), and Grit (Ma et al., 2023). Our PST outperforms all baselines on the PascalVOC-SP and Peptides-Func datasets. Additionally, it attained the second-highest performance on the Peptides-Struct dataset. This demonstrates its remarkable ability to capture long-range interactions and produce competitive results across various benchmark datasets.

Table 4: Results on long range graph benchmark. * stands for using RWSE, ** stands for using LapPE. All baselines are around 500k parameters. PST takes about 500k parameters on PascalVOC-SP and about 1M on the other two datasets.

| Model | PascalVOC-SP F1 score ↑ | Peptides-Func AP ↑ | Peptides-Struct MAE ↓ |
|---|---|---|---|
| GCN | $0.1268_{\pm0.0060}$ | $0.5930_{\pm0.0023}$ | $0.3496_{\pm0.0013}$ |
| GINE | $0.1265_{\pm0.0076}$ | $0.5498_{\pm0.0079}$ | $0.3547_{\pm0.0045}$ |
| GatedGCN | $0.2873_{\pm0.0219}$ | $0.5864_{\pm0.0077}$ | $0.3420_{\pm0.0013}$ |
| GatedGCN* | $0.2860_{\pm0.0085}$ | $0.6069_{\pm0.0035}$ | $0.3357_{\pm0.0006}$ |
| Transformer** | $0.2694_{\pm0.0098}$ | $0.6326_{\pm0.0126}$ | $0.2529_{\pm0.0016}$ |
| SAN* | $0.3216_{\pm0.0027}$ | $0.6439_{\pm0.0075}$ | $0.2545_{\pm0.0012}$ |
| SAN** | $0.3230_{\pm0.0039}$ | $0.6384_{\pm0.0121}$ | $0.2683_{\pm0.0043}$ |
| GraphGPS | $0.3748_{\pm0.0109}$ | $0.6535_{\pm0.0041}$ | $0.2500_{\pm0.0005}$ |
| Exphormer | $0.3975_{\pm0.0037}$ | $0.6527_{\pm0.0043}$ | $0.2481_{\pm0.0007}$ |
| GMLP-Mixer | - | $0.6970_{\pm0.0080}$ | $0.2475_{\pm0.0015}$ |
| Graph ViT | - | $0.6942_{\pm0.0075}$ | $\mathbf{0.2449_{\pm0.0016}}$ |
| Grit | - | $\mathbf{0.6988_{\pm0.0082}}$ | $0.2460_{\pm0.0012}$ |
| PST | $\mathbf{0.4010_{\pm0.0072}}$ | $\mathbf{0.6984_{\pm0.0051}}$ | $0.2470_{\pm0.0015}$ |

## 8 CONCLUSION

We introduce a novel approach employing symmetric rank decomposition to transform interconnected nodes within a graph into independent nodes with associated coordinates. Additionally, we propose the Point Set Transformer (PST) to encode the point set. Our approach demonstrates remarkable theoretical expressivity and excels in real-world performance, addressing both short-range and long-range tasks effectively. It also provides a new paradigm for graph machine learning.

## 9 LIMITATIONS

Our models' scalability is still constrained by the Transformer architecture. To overcome this, acceleration techniques such as sparse attention and linear attention could be explored, which will be our future work.

## 10 REPRODUCIBILITY STATEMENT

Our code is in the supplementary material. Proofs of all theorems in the maintext are in Appendix A.

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

## A  PROOF

### A.1  PROOF OF PROPOSITION 1

The matrices $Q_1^T Q_1$ and $Q_2^T Q_2$ in $\mathbb{R}^{r \times r}$ are full rank and thus invertible. This allows us to derive the following equations:

$$L = Q_1 Q_1^T = Q_2 Q_2^T \tag{8}$$

$$Q_1 Q_1^T = Q_2 Q_2^T \Rightarrow Q_1^T Q_1 Q_1^T = Q_1^T Q_2 Q_2^T \tag{9}$$

$$\Rightarrow Q_1^T = (Q_1^T Q_1)^{-1} Q_1^T Q_2 Q_2^T \tag{10}$$

$$\Rightarrow \exists R \in \mathbb{R}^{m \times m}, Q_1^T = R Q_2^T \tag{11}$$

$$\Rightarrow \exists R \in \mathbb{R}^{m \times m}, Q_1 = Q_2 W \tag{12}$$

$$Q_1 Q_1^T = Q_2 R R^T Q_2^T = Q_2 Q_2^T \Rightarrow Q_2^T Q_2 R R^T Q_2^T Q_2 = Q_2^T Q_2 Q_2^T Q_2 \tag{13}$$

$$\Rightarrow R R^T = (Q_2^T Q_2)^{-1} Q_2^T Q_2 Q_2^T Q_2 (Q_2^T Q_2)^{-1} = I \tag{14}$$

Since $R$ is orthogonal, any two full rank $Q$ matrices are connected by an orthogonal transformation. Furthermore, if there exists an orthogonal matrix $R$ where $R R^T = I$, then $Q_1 = Q_2 R$, and $L = Q_1 Q_1^T = Q_2 R R^T Q_2^T = Q_2 Q_2^T$.

## A.2 MATRIX D+A IS ALWAYS POSITIVE SEMI-DEFINITE

$\forall x \in \mathbb{R}^n$,

$$x^T(D + A)x = \sum_{(i,j) \in E} x_i x_j + \sum_{i \in V} (\sum_{j \in V} A_{ij}) x_i^2 \tag{15}$$

$$= \sum_{(i,j) \in E} x_i x_j + \frac{1}{2} \sum_{(i,j) \in E} x_i^2 + \frac{1}{2} \sum_{(i,j) \in E} x_j^2 \tag{16}$$

$$= \frac{1}{2} \sum_{(i,j) \in E} (x_i + x_j)^2 \geq 0 \tag{17}$$

Therefore, $D + A$ is always positive semi-definite.

## A.3 PROOF OF THEOREM 1

We restate the theorem here:

**Theorem 8.** *Given two graphs $\mathcal{G} = (V, A, X)$ and $\mathcal{G}' = (V', A', X')$ with degree matrices $D$ and $D'$, respectively, the two graphs are isomorphic ($\mathcal{G} \simeq \mathcal{G}'$) if and only if $\exists R \in O(r), \{\!\{(X_v, RQ_v) | \forall v \in V\}\!\} = \{\!\{(X_v, Q'_v) | v \in V'\}\!\}$, where $r$ denotes the rank of matrix $A$, and $Q$ and $Q'$ are the symmetric rank decompositions of $D + A$ and $D' + A'$ respectively.*

*Proof.* Two graphs are isomorphic $\Leftrightarrow \exists \pi \in \Pi_n, \pi(A) = A'$ and $\pi(X) = X'$.

Now we prove that $\exists \pi \in \Pi_n, \pi(A) = A'$ and $\pi(X) = X \Leftrightarrow \exists R \in O(r), \{\!\{(X_v, RQ_v) | v \in V\}\!\} = \{\!\{(X_v, Q'_v) | v \in V'\}\!\}$.

When $\exists \pi \in \Pi_n, \pi(A) = A'$ and $\pi(X) = X'$, as

$$\pi(Q)\pi(Q)^T = \pi(A + D) = A' + D' = Q'Q'^T, \tag{18}$$

according to Proposition 1, $\exists R \in O(r), \pi(Q)R^T = Q'$. Moreover, $\pi(X) = X'$, so

$$\{\!\{(X_v, RQ_v) | v \in V\}\!\} = \{\!\{(X'_v, Q'_v) | v \in V'\}\!\} \tag{19}$$

When $\exists R \in O(r), \{\!\{(X_v, RQ_v) | v \in V\}\!\} = \{\!\{(X'_v, Q'_v) | v \in V'\}\!\}$, there exists permutation $\pi \in \Pi_n$, $\pi(X) = X', \pi(Q)R^T = Q'$. Therefore,

$$\pi(A + D) = \pi(Q)\pi(Q)^T = \pi(Q)R^T R\pi(Q)^T = Q'Q'^T = A' + D' \tag{20}$$

$\square$

As $A = D + A - \frac{1}{2}\text{diag}((D + A)\vec{1}), A' = D + A - \frac{1}{2}\text{diag}((D + A)\vec{1})$, where $\vec{1} \in \mathbb{R}^n$ is an vector with all elements $= 1$.

$$\pi(A) = A' \tag{21}$$

## A.4 PROOF OF THEOREM 2

Now we restate the theorem.

**Theorem 9.** *Given two graphs $\mathcal{G} = (V, A, X)$ and $\mathcal{G} = (V', A', X')$, and an injective permutation-equivariant function $Z$ mapping adjacency matrix to a symmetric matrix: (1) For all permutation-equivariant function $f$, if $\mathcal{G} \simeq \mathcal{G}'$, then the two sets of generalized coordinates are equal up to an orthogonal transformation, i.e., $\exists R \in O(r), \{\!\{X_v, R\tilde{Q}(Z(A), f)_v | v \in V\}\!\} = \{\!\{X'_v, \tilde{Q}(Z(A'), f)_v | v \in V'\}\!\}$, where $r$ is the rank of $A$, $\tilde{Q}, \tilde{Q}'$ are the generalized coordinates of $A$ and $A'$ respectively. (2) There exists a continuous permutation-equivariant function $f : \mathbb{R}^r \to \mathbb{R}^{r \times 2}$, such that $\mathcal{G} \simeq \mathcal{G}'$ if $\exists R \in O(r), \forall i = 1, 2, ..., d, \{\!\{(X_v, R\tilde{Q}(Z(A), f_1)_v, R\tilde{Q}(Z(A), f_2)_v) | v \in V\}\!\} = \{\!\{(X'_v, \tilde{Q}(Z(A'), f_1)_v, \tilde{Q}(Z(A'), f_2)_v) | v \in V'\}\!\}$, where $f_1 : \mathbb{R}^r \to \mathbb{R}^r$ and $f_2 : \mathbb{R}^r \to \mathbb{R}^r$ are two output channels of $f$.*

*Proof.* First, as $Z$ is an injective function, forall permutation $\pi \in \Pi_n$

$$\pi(Z(A)) = Z(A') \Leftrightarrow Z(\pi(A)) = Z(A') \Leftrightarrow \pi(A) = A'. \tag{22}$$

Therefore, two matrix are isomorphic $\Leftrightarrow \exists \pi \in \Pi_n, \pi(X) = X', \pi(Z) = Z'$, where $Z, Z'$ denote $Z(A), Z(A')$ respectively.

In this proof, we denote eigendecomposition as $Z = U\mathrm{diag}(\Lambda)U^T$ and $Z' = U'\mathrm{diag}(\Lambda')U'^T$, where elements in $\Lambda$ and $\Lambda'$ are sorted in ascending order. Let the multiplicity of eigenvalues in $Z$ be $r_1, r_2, ..., r_l$, corresponding to eigenvalues $\lambda_1, \lambda_2, ..., \lambda_i$.

(1) If $\mathcal{G} \simeq \mathcal{G}'$, there exists a permutation $\pi \in \Pi_n, \pi(X) = X', \pi(Z) = Z'$.

$$\pi(Z) = Z' \Rightarrow Z' = \pi(U)\mathrm{diag}(\Lambda)\pi(U)^T = U'\mathrm{diag}(\Lambda')U'^T. \tag{23}$$

$\pi(U)\mathrm{diag}(\Lambda)\pi(U)^T$ is also an eigendecomposition of $Z'$, so $\Lambda = \Lambda'$ as they are both sorted in ascending order. Moreover, since $\pi(U), U'$ are both matrices of eigenvectors, they can differ only in the choice of bases in each eigensubspace. So there exists a block diagonal matrix $V$ with orthogonal matrix $V_1 \in O(r_1), V_2 \in O(r_2), ..., V_l \in O(r_l)$ as diagonal blocks that $\pi(U)V = U'$.

As $f$ is a permutation equivariant function,

$$\Lambda_i = \Lambda_j \Rightarrow \exists \pi \in \Pi_r, \pi(i) = j, \pi(j = i), \pi(\Lambda) = \Lambda \tag{24}$$
$$\Rightarrow \exists \pi \in \Pi_r, \pi(i) = j, \pi(j = i), \pi(f(\Lambda)) = f(\pi(\Lambda)) = f(\Lambda) \tag{25}$$
$$\Rightarrow f(\Lambda)_i = f(\Lambda)_j \tag{26}$$

Therefore, $f$ will produce the same value on positions with the same eigenvalue. Therefore, $f$ can be consider as a block diagonal matrix with $f_1 I_{r_1}, f_2 I_{r_2}, ..., f_l I_{r_l}$ as diagonal blocks, where $f_i \in \mathbb{R}$ is $f(\Lambda)_{p_i}$, $p_i$ is a position that $\Lambda_{p_i} = \lambda_i$, and $I_r$ is an identity matrix $\in \mathbb{R}^{r \times r}$.

Therefore,

$$V\mathrm{diag}(f(\Lambda)) = \mathrm{diag}(f_1 V_1, f_2 V_2, ..., f_l V_l) = \mathrm{diag}(f(\Lambda))V. \tag{27}$$

Therefore,

$$\pi(Q(Z, f))V = \pi(U)\mathrm{diag}(f(\Lambda))V \tag{28}$$
$$= \pi(U)V\mathrm{diag}(f(\Lambda)) \tag{29}$$
$$= U'\mathrm{diag}(f(\Lambda')) \tag{30}$$
$$= Q(Z', f) \tag{31}$$

As $VV^T = I, V \in O(r)$,

$$\exists R \in O(r), \{\!\{X_v, R\tilde{Q}(Z(A), f)_v | v \in V\}\!\} = \{\!\{X_v', \tilde{Q}(Z(A'), f)_v | v \in V'\}\!\} \tag{32}$$

(2) We simply define $f_1$ is element-wise abstract value and square root $\sqrt{|.|}$, $f_1$ is element-wise abstract value and square root multiplied with its sign $sgn(|.|)\sqrt{|.|}$. Therefore, $f_1, f_2$ are continuous and permutation equivariant.

if $\exists R \in O(r)$,

$\{\!\{X_v, R\tilde{Q}(Z(A), f_1)_v, R\tilde{Q}(Z(A), f_2)_v | v \in V\}\!\} = \{\!\{X_v', \tilde{Q}(Z(A'), f_1)_v, \tilde{Q}(Z(A'), f_2)_v | v \in V'\}\!\}$.
then there exist $\pi \in \Pi_n$, so that

$$\pi(X) = X' \tag{33}$$
$$\pi(U)\mathrm{diag}(f_1(\Lambda))R^T = U'\mathrm{diag}(f_1(\Lambda')) \tag{34}$$
$$\pi(U)\mathrm{diag}(f_2(\Lambda))R^T = U'\mathrm{diag}(f_2(\Lambda')). \tag{35}$$

Therefore,

$$\pi(Z) = \pi(U)\mathrm{diag}(f_1(\Lambda))\mathrm{diag}(f_2(\Lambda))\pi(U')^T \tag{36}$$
$$= \pi(U)\mathrm{diag}(f_1(\Lambda))RR^T\mathrm{diag}(f_2(\Lambda))\pi(U')^T \tag{37}$$
$$= U'\mathrm{diag}(f_1(\Lambda'))\mathrm{diag}(f_2(\Lambda'))U'^T \tag{38}$$
$$= Z'. \tag{39}$$

As $\pi(Z) = Z', \pi(X) = X'$, two graphs are isomorphic.

$\square$

## A.5 PROOF OF THEOREM 5

Let $H_l$ denote a circle of $l$ nodes. Let $G_l$ denote a graph of two connected components, one is $H_{\lfloor l/2 \rfloor}$ and the other is $H_{\lceil l/2 \rceil}$. Obviously, there exists a node pair in $G_l$ with shortest path distance equals to infinity, while $H_l$ does not have such a node pair. So the multiset of shortest path distance is easy to distinguish them. However, they are regular graphs with node degree all equals to 2, so MPNN cannot distinguish them:

**Lemma 1.** *For all nodes $v, u$ in $G_l$, $H_l$, they have the same representation produced by $k$-layer MPNN, forall $k \in N$.*

*Proof.* We proof it by induction.

$k = 0$. Initialization, all node with trivial node feature and are the same.

Assume $k - 1$-layer MPNN still produce representation $h$ for all node. At the $k$-th layer, each node's representation will be updated with its own representation and two neighbors representations as follows.

$$h, \{\!\{h, h\}\!\} \tag{40}$$

So all nodes still have the same representation. $\square$

## A.6 PROOF OF THEOREM 4 AND 14

Given two function $f, g$, $f$ can be expressed by $g$ means that there exists a function $\phi$ $\phi \circ g = f$, which is equivalent to given arbitrary input $H, G$, $f(H) = f(G) \Rightarrow g(H) = g(G)$. We use $f \to g$ to denote that $f$ can be expressed with $g$. If both $f \to g$ and $g \to f$, there exists a bijective mapping between the output of $f$ to the output of $g$, denoted as $f \leftrightarrow g$.

Here are some basic rule.

- $g \to h \Rightarrow f \circ g \to f \circ h$.
- $g \to h, f \to s \Rightarrow f \circ g \to s \circ h$.
- $f$ is bijective, $f \circ g \to g$

2-folklore Weisfeiler-Leman test produce a color $h_{ij}^t$ for each node pair $(i, j)$ at $t$-th iteration. It updates the color as follows,

$$h_{ij}^{t+1} = \text{hash}(h_{ij}^t, \{\!\{(h_{ik}^t, h_{kj}^t)|k \in V\}\!\}). \tag{41}$$

The color of the the whole graph is

$$h_G^t = \text{hash}(\{\!\{h_{ij}^t|(i, j) \in V \times V\}\!\}). \tag{42}$$

Initially, tuple color hashes the node feature and edge between the node pair, $h_{ij}^0 \to \delta_{ij}, A_{ij}, X_i, X_j$.

We are going to prove that

**Lemma 2.** *Forall $t \in \mathbb{N}$, $h_{ij}^t$ can express $A_{ij}^k$, $k = 0, 1, 2, ..., 2^t$, where $A$ is the adjacency matrix of the input graph.*

*Proof.* We prove it by induction on $t$.

- When $t = 0$, $h_{ij}^0 \to A_{ij}, I_{ij}$ in initialization.

- If $t > 0, \forall t' < t, h_{ij}^{t'} \to A^{k'}, k' = 0, 1, 2, ..., 2^{t'}$. For all $k = 0, 1, 2$,

$$h_{ij}^t \to \text{hash}(h_{ij}^t, \{\!\{(h_{ik}^t, h_{kj}^t)|k \in V\}\!\}) \tag{43}$$

$$\to \text{hash}(h_{ij}^t, \{\!\{(A_{ik}^{\lfloor k/2 \rfloor}, A_{kj}^{\lceil k/2 \rceil})|k \in V\}\!\}) \tag{44}$$

$$\to \sum_{k \in V} A_{ik}^{\lfloor k/2 \rfloor} A_{kj}^{\lceil k/2 \rceil} \tag{45}$$

$$\to A_{ij}^k \tag{46}$$

$\square$

To prove that $t$-iteration 2-FWL cannot compute shortest path distance larger than $2^{t'}$, we are going to construct an example.

**Lemma 3.** *Let $H_l$ denote a circle of $l$ nodes. Let $G_l$ denote a graph of two connected components, one is $H_{\lfloor l/2 \rfloor}$ and the other is $H_{\lceil l/2 \rceil}$. $\forall K \in \mathbb{N}^+$, 2-FWL can not distinguish $H_{l_K}$ and $G_{l_K}$, where $l_K = 2 \times 2 \times (2^K)$. However, $G_{l_K}$ contains node tuple with $2^K + 1$ shortest path distance between them while $H_{l_K}$ does not, any model count up to $2^K + 1$ shortest path distance can count it.*

*Proof.* Given a fixed $t$, we enumerate the iterations of 2-FWL. Given two graphs $H_{l_K}$, $G_{l_K}$, we partition all tuples in each graph according to the shortest path distance between nodes: $c_0, c_1, ..., c_l, ..., c_{2^K}$, where $c_l$ contains all tuples with shortest path distance between them as $l$, and $c_{>2^K}$ contains all tuples with shortest path distance between them larger than $2^K$. We are going to prove that at $k$-th layer $k <= K$, all $c_i, i \leq 2^k$ nodes have the same representation (denoted as $h_i^k$) $c_{2^k+1}, c_{2^k+2}, ..., c_{2^K}, c_{>K}$ nodes all have the same representation (denoted as $h_{2^k+1}^k$).

Initially, all $c_0$ tuples have representation $h_0^0$, all $c_1$ tuples have the same representation $h_1^0$ in both graph, and all other tuples have the same representation $h_2^0$.

Assume at $k$-th layer, all $c_i, i \leq 2^k$ nodes have the same representation $h_i^k$, $c_{2^k+1}, c_{2^k+2}, ..., c_{2^K}, c_{>2^K}$ tuples all have the same representation $h_{2^k+1}^k$. At $k+1$-th layer, each representation is updated as follows.

$$h_{ij}^{t+1} \leftarrow h_{ij}^t, \{\!\{(h_{iv}^t, h_{vj}^t)|v \in V\}\!\} \tag{47}$$

For all tuples, the multiset has $l_K$ elements in total.

For $c_0$ tuples, the multiset have 1 $(h_0^k, h_0^k)$ as $v = i$, 2 $(h_t^k, h_t^k)$ for $t = 1, 2, .., 2^k$ respectively as v is the $k$-hop neighbor of $i$, and all elements left are $(h_{2^k+1}^k, h_{2^k+1}^k)$ as $v$ is not in the $k$-hop neighbor of $i$.

For $c_t, t = 1, 2, ..., 2^k$ tuples: the multiset have 1 $(h_a^k, h_{t-a}^k)$ for $a = 0, 1, 2, .., t$ respectively as $v$ is on the shortest path between $(i, j)$, and 1 $(h_a^k, h_{2^k+1}^k)$ for $a = 1, 2, ..., 2^k$ respectively, and 1 $(h_{2^k+1}^k, h_a^k)$ for $a = 1, 2, ..., 2^k$ respectively, with other elements are $(h_{2^k+1}^k, h_{2^k+1}^k)$.

For $c_t, t = 2^k+1, 2^k+2, ..., 2^{k+1}$ tuples: the multiset have 1 $(h_a^k, h_{t-a}^k)$ for $a = t - 2^k, t - 2^k + 1, ..., 2^k$ respectively as $v$ is on the shortest path between $(i, j)$, 1 $(h_a^k, h_{t-a}^k)$ for $a \in \{0, 1, 2, ..., t - 2^k - 1\} \cup \{2^k + 1, 2^k + 2, ..., 2^{k+1}\}$ respectively as $v$ is on the shortest path between $(i, j)$, and 1 $(h_a^k, h_{2^k+1}^k)$ for $a = 1, 2, ..., 2^k$ respectively, and 1 $(h_{2^k+1}^k, h_a^k)$ for $a = 1, 2, ..., 2^k$ respectively, with other elements are $(h_{2^k+1}^k, h_{2^k+1}^k)$.

For $c_t, t = 2^{k+1} + 1, ..., 2^K, > 2^K$: the multiset are all the same : 2 $(h_a^k, h_{2^k+1}^k)$ and 2 $(h_{2^k+1}^k, h_a^k)$ for $a = 1, 2, 3, ..., 2^k$, respectively. $\square$

## A.7 PROOF OF THEOREM 3

We can simply choose $f_k(\Lambda) = \Lambda^k$. Then $\langle \tilde{Q}(A, f_0)_i, \tilde{Q}(A, f_k)_j \rangle = A_{ij}^k$. The shortest path distance is

$$spd(i, j, A) = \arg\min_k \{k \in \mathbb{N} | A_{ij}^k > 0\} \tag{48}$$

## A.8 PROOF OF THEOREM 6 AND 7

This section assumes that the input graph is undirected and unweighted with no self-loops. Let $A$ denote the adjacency matrix of the graph. Note that $A^T = A, A \odot A = A$

An $L$-path is a sequence of edges $[(i_1, i_2), (i_2, i_3), ..., (i_L, i_{L+1})]$, where all nodes are different from each other. An $L$-cycle is an $L$-path except that $i_1 = i_{L+1}$. Two paths/cycles are considered equivalent if their sets of edges are equal. The count of $L$ path from node $u$ to $v$ is the number of

non-equivalent pathes with $i_1 = u, i_{L+1} = v$. The count of $L$-cycle rooted in node $u$ is the number of non-equivalent cycles involves node $u$.

Perepechko & Voropaev (2009) show that the number of path can be expressive with a polynomial of $A$, where $A$ is the adjacency matrix of the input unweight graph. Specifically, let $P_L$ denote path matrix whose $(u, v)$ elements denote the number of $L$-pathes from $u$ to $v$, Perepechko & Voropaev (2009) provides formula to express $P_L$ with $A$ for small $L$.

This section considers a weaken version of point cloud transformer. Each layer still consists of sv-mixer and multi-head attention. However, the multi-head attention matrix takes the scalar and vector feature before sv-mixer for $Q, K$ and use the feature after sv-mixer for $V$.

At $k$-th layer sv-mixer:

$$s_i' \leftarrow \text{MLP}_1(s_i \| \text{diag}(W_1 v_i v_i^{k,T} W_2)) \tag{49}$$

$$v_i' \leftarrow W_3 \text{diag}(\text{MLP}_2(s_i')) v_i + W_4 v_i \tag{50}$$

Multi-head Attention:

$$Y_{ij} = \text{MLP}_3(K_{ij}), K_{ij} = (W_q^s s_i \odot W_k^s s_j) \| \text{diag}(W_q^v v_i v_i^T W_k^v), \tag{51}$$

As $s_i'$ and $v_i'$ can express $s_i, v_i$, so the weaken version can be expressed with the original version.

$$s_i \leftarrow \text{MLP}_4(s_j' \| \sum_j \text{Atten}_{ij} s_j') \tag{52}$$

$$v_i \leftarrow W_5(v_j' \| \sum_j \text{Atten}_{ij} v_j') \tag{53}$$

Let $Y^k$ denote the attention matrix at $k$-th layer. $Y^k$ is a learnable function of $A$. Let $\mathbb{Y}^k$ denote the polynomial space of $A$ that $Y^k$ can express. Each element in it is a function from $\mathbb{R}^{n \times n} \to \mathbb{R}^{n \times n}$

We are going to prove some lemmas about $\mathbb{Y}$.

**Lemma 4.** $\mathbb{Y}^k \subseteq \mathbb{Y}^{k+1}$

*Proof.* As there exists residual connection, scalar and vector representations of layer $k + 1$ can always contain those of layer $k$, so attention matrix of layer $k + 1$ can always express those of layer $k$. $\square$

**Lemma 5.** *If $y_1, y_2, ..., y_s \in \mathbb{Y}^k$, their hadamard product $y_1 \odot y_2 \odot ... \odot y_s \in \mathbb{Y}^k$.*

*Proof.* As $(y_1 \odot y_2 \odot ... \odot y_s)_{ij} = \prod_{l=1}^s (y_l)_{ij}$ is a element-wise polynomial on compact domain, an MLP (denoted as $g$) exists that takes $(i, j)$ elements of the $y_1, y_2, ..., y_s$ to produce the corresponding elements of their hadamard product. Assume $g_0$ is the MLP$_3$ in Equation 52 to produce the concatenation of $y_1, y_2, .., y_s$, use $g \circ g_0$ (the composition of two mlps) for the MLP$_3$ in Equation 52 produces the hadamard product. $\square$

**Lemma 6.** *If $y_1, y_2, ..., y_s \in \mathbb{Y}^k$, their linear combination $\sum_{l=1}^s a_l y_l \in \mathbb{Y}^k$, where $a_l \in \mathbb{R}$.*

*Proof.* As $(\sum_{l=1}^s a_l y_l)_{ij} = \sum_{l=1}^s a_l (y_l)_{ij}$ is a element-wise linear layer (denoted as $g$). Assume $g_0$ is the MLP$_3$ in Equation 52 to produce the concatenation of $y_1, y_2, .., y_s$, use $g \circ g_0$ for the MLP$_3$ in Equation 52 produces the linear combination. $\square$

**Lemma 7.** $\forall s > 0, A^s \in \mathbb{Y}^1$.

*Proof.* As shown in Section 5.1, the inner product of coordinates can produce $A^s$. $\square$

**Lemma 8.** $\forall y_1, y_2, y_3 \in \mathbb{Y}^k, s \in \mathbb{N}^+, d(y_1)y_2, y_2 d(y_1), d(y_1)y_2 d(y_3), y_1 A^s, A^s y_1, y_1 A^s y_2 \in \mathbb{Y}^k$

*Proof.* According to Equation 49 and Equation 51, $s'_i$ at $k$-th layer can express $y_{ii}$ for all $y \in \mathbb{Y}^k$. Therefore, at $k + 1$-th layer in Equation 51, MLP$_3$ can first compute element $(i, j)$ $(y_2)_{ij}$ from $s_i, s_j, v_i, v_j$, then multiply $(y_2)_{ij}$ with $(y_1)_{ii}$ from $s_i$, $(y_3)_{jj}$ from $s_j$ and thus produce $d(y_1)y_2, y_2 d(y_1), d(y_1)y_2 d(y_3)$.

Moreover, according to Equation 52, $v_i$ at $k + 1$-th layer can express $\sum_k (y_1)_{ik} v_k, \sum_k (y_2)_{ik} v_k$. So at $k + 1$-th layer, the $(i, j)$ element can express $\langle \sum_k (y_1)_{ik} v_k, v_j \rangle, \langle v_i, \sum_k (y_1)_{jk} v_k \rangle, \langle (y_1)_{ik} v_k, \sum_k (y_2)_{jk} v_k \rangle$, corresponds to $y_1 A^s, A^s y_2, y_1 A^s y_2$, respectively.

$\square$

Therefore,

**Lemma 9.**
- $\forall s > 0, l > 0, a_i > 0, \odot_{i=1}^l A^{a_i} \in \mathbb{Y}^1$.

- $\forall s_1, s_2 > 0, l > 0, A^{s_1} d(\odot_{i=1}^l A^{a_i}), d(\odot_{i=1}^l A^{a_i}) A^{s_1}, d(\odot_{i=1}^{l_1} A^{b_i}) A^{s_1} d(\odot_{i=1}^{l_2} A^{b_i}) \in \mathbb{Y}^2$.

- $\forall s_1, s_2, s_3 > 0, A^{s_1} d(\odot_{i=1}^l A^{a_i})$

Therefore, we come to our main theorem.

**Theorem 10.** *The attention matrix of 1-layer PST can express $P_2$, 2-layer PST can express $P_3$, 3-layer PST can express $P_4, P_5$, 5-layer PST can express $P_6$.*

*Proof.* As shown in (Perepechko & Voropaev, 2009),

$$P_2 = A^2 \tag{54}$$

Only one kind basis $\odot_{i=1}^l A^{a_i}$. 1-layer PST can express it.

$$P_3 = A^3 + A - Ad(A^2) - d(A^2)A \tag{55}$$

Three kind of basis $\odot_{i=1}^l A^{a_i}(A^3, A)$, $A^{s_1} d(\odot_{i=1}^l A^{a_i})(Ad(A^2))$, and $d(\odot_{i=1}^l A^{a_i}) A^{s_1}$. 2-layer PST can express it.

$$P_4 = A^4 + A^2 + 3A \odot A^2 - d(A^3)A - d(A^2)A^2 - Ad(A^3) - A^2 d(A^2) - Ad(A^2)A \tag{56}$$

Four kinds of basis $\odot_{i=1}^l A^{a_i}$ $(A^4, A^2, A \odot A^2)$, $A^{s_1} d(\odot_{i=1}^l A^{a_i})$ $(Ad(A^3), A^2 d(A^2))$, $d(\odot_{i=1}^l A^{a_i}) A^{s_1}$ $(d(A^3)A, d(A^2)A^2)$, and $A^{s_1} d(\odot_{i=1}^l A^{a_i}) A^{s_3}$ $(Ad(A^2)A)$. 3-layer PST can express it.

$$P_5 = A^5 + 3A^3 + 4A \tag{57}$$
$$+ 3A^2 \odot A^2 \odot A + 3A \odot A^3 - 4A \odot A^2 \tag{58}$$
$$- d(A^4)A - d(A^3)A^2 - d(A^2)A^3 + 2d(A^2)^2 A - 2d(A^2)A - 4d(A^2)A \tag{59}$$
$$- Ad(A^4) - A^2 d(A^3) - A^3 d(A^2) + 2Ad(A^2)^2 - 2Ad(A^2) - 4Ad(A^2) \tag{60}$$
$$+ d(A^2)Ad(A^2) \tag{61}$$
$$+ 3(A \odot A^2)A \tag{62}$$
$$+ 3A(A \odot A^2) \tag{63}$$
$$- Ad(A^3)A - Ad(A^2)A^2 - A^2 d(A^2)A \tag{64}$$
$$+ d(Ad(A^2)A)A \tag{65}$$
$$\tag{66}$$

Basis are in

- $\mathbb{Y}^1$

- $\odot_{i=1}^{l} A^{a_i}$: $A^5, A^3, A, A^2 \odot A^2 \odot A, A \odot A^3, A \odot A^2$.

- $\mathbb{Y}^2$

  - $A^{s_1} d(\odot_{i=1}^{l} A^{a_i})$: $Ad(A^4), A^2 d(A^3), A^3 d(A^2), Ad(A^2)^2, Ad(A^2), Ad(A^2)$.
  - $d(\odot_{i=1}^{l} A^{a_i}) A^{s_1}$: $Ad(A^4), A^2 d(A^3), A^3 d(A^2), Ad(A^2)^2, Ad(A^2), Ad(A^2)$.
  - $d(\odot_{i=1}^{l_1} A^{a_i}) A^{s_1} d(\odot_{i=1}^{l_1} A^{b_i})$: $d(A^2) Ad(A^2)$.
  - $A^{s_1} (\odot_{i=1}^{l} A^{a_i})$: $A(A \odot A^2)$.
  - $(\odot_{i=1}^{l} A^{a_i}) A^{s_1}$: $(A \odot A^2) A$

- $\mathbb{Y}^3$:

  - $A^s \mathbb{Y}^2$: $Ad(A^3)A, Ad(A^2)A^2, A^2 d(A^2)A$.
  - $d(\mathbb{Y}^2) A^s$: $d(Ad(A^2)A)A$

3-layer PST can express it.

Formula for 6-path matrix is quite long.

$$P_6 = A^6 + 4A^4 + 12A^2 \tag{67}$$
$$+ 3A \odot A^4 + 6A \odot A^2 \odot A^3 + A^2 \odot A^2 \odot A^2 - 4A^2 \odot A^2 + 44A \odot A^2 \tag{68}$$
$$- d(A^5)A - d(A^4)A^2 - d(A^3)A^3 - 5d(A^3)A - d(A^2)A^4 - 7d(A^2)A^2 \tag{69}$$
$$+ 2d(A^2)^2 A^2 + 4(d(A^2) \odot d(A^3))A \tag{70}$$
$$- Ad(A^5) - A^4 d(A^2) - A^3 d(A^3) - 5Ad(A^3) - A^2 d(A^4) - 7A^2 d(A^2) \tag{71}$$
$$+ 2A^2 d(A^2)^2 + 4A(d(A^2) \odot d(A^3)) \tag{72}$$
$$+ d(A^2)Ad(A^3) + d(A^3)Ad(A^2) + d(A^2)A^2 d(A^2) \tag{73}$$
$$+ 2(A \odot A^3)A + 2(A \odot A^2 \odot A^2)A + (A^2 \odot A^2 \odot A)A - 3(A \odot A^2)A + (A \odot A^3)A \tag{74}$$
$$+ (A \odot A^2)A^2 + 2(A \odot A^2)A^2 - (A \odot A^2)A \tag{75}$$
$$+ 2A(A \odot A^3) + 2A(A \odot A^2 \odot A^2) + A(A^2 \odot A^2 \odot A) - 3A(A \odot A^2) + A(A \odot A^3) \tag{76}$$
$$+ A^2(A \odot A^2) + 2A^2(A \odot A^2) - A(A \odot A^2) \tag{77}$$
$$- 8A \odot (A(A \odot A^2)) - 8A \odot ((A \odot A^2)A) \tag{78}$$
$$- 12d(A^2)(A \odot A^2) - 12(A \odot A^2)d(A^2) \tag{79}$$
$$- Ad(A^4)A - Ad(A^2)A^3 - A^3 d(A^2)A - Ad(A^3)A^2 - A^2 d(A^3)A - A^2 d(A^2)A^2 \tag{80}$$
$$- 10Ad(A^2)A + 2Ad(A^2)^2 A \tag{81}$$
$$+ d(A^2)Ad(A^2)A + Ad(A^2)Ad(A^2) \tag{82}$$
$$- 3A \odot (Ad(A^2)A) \tag{83}$$
$$- 4Ad((A \odot A^2)A) - 4Ad(A(A \odot A^2)) \tag{84}$$
$$+ 3A(A \odot A^2)A \tag{85}$$
$$- 4d(A(A \odot A^2))A - 4d((A \odot A^2)A)A \tag{86}$$
$$+ d(Ad(A^3)A)A + d(Ad(A^2)A)A^2 + d(Ad(A^2)A^2)A + d(A^2 d(A^2)A)A \tag{87}$$
$$+ Ad(Ad(A^3)A) + A^2 d(Ad(A^2)A) + Ad(A^2 d(A^2)A) + Ad(Ad(A^2)A^2) \tag{88}$$
$$+ Ad(Ad(A^2)A)A \tag{89}$$
$$\tag{90}$$

Basis are in

- $\mathbb{Y}^1$

- $\odot_{i=1}^{l} A^{a_i}$: $A^6, A^4, A^2, A \odot A^4, A \odot A^2 \odot A^3, A^2 \odot A^2 \odot A^2, A^2 \odot A^2, A \odot A^2$.

- $\mathbb{Y}^2$

  - $A^{s_1} d(\odot_{i=1}^{l} A^{a_i})$: $Ad(A^5)$, $A^4 d(A^2)$, $A^3 d(A^3)$, $Ad(A^3)$, $A^2 d(A^4)$, $A^2 d(A^2)$, $A^2 d(A^2)^2$, $A(d(A^2) \odot d(A^3))$.
  - $d(\odot_{i=1}^{l} A^{a_i}) A^{s_1}$: $d(A^5)A$, $d(A^4)A^2$, $d(A^3)A^3$, $d(A^3)A$, $d(A^2)A^4$, $d(A^2)A^2$, $d(A^2)^2 A^2$, $(d(A^2) \odot d(A^3))A$.
  - $d(\odot_{i=1}^{l_1} A^{a_i}) A^{s_1} d(\odot_{i=1}^{l_1} A^{b_i})$: $d(A^2) Ad(A^3)$, $d(A^3) Ad(A^2)$, $d(A^2) A^2 d(A^2)$.
  - $A^{s_1} (\odot_{i=1}^{l} A^{a_i})$: $A(A \odot A^3)$, $A(A \odot A^2 \odot A^2)$, $A(A^2 \odot A^2 \odot A)$, $A(A \odot A^2)$, $A(A \odot A^3)$, $A^2(A \odot A^2)$, $A^2(A \odot A^2)$, $A(A \odot A^2)$.
  - $(\odot_{i=1}^{l} A^{a_i}) A^{s_1}$: $(A \odot A^3)A$, $(A \odot A^2 \odot A^2)A$, $(A^2 \odot A^2 \odot A)A$, $(A \odot A^2)A$, $(A \odot A^3)A$, $(A \odot A^2)A^2$, $(A \odot A^2)A^2$, $(A \odot A^2)A$
  - $\mathbb{Y}^2 \odot \mathbb{Y}^2$: $A \odot ((A \odot A^2)A)$, $A \odot ((A \odot A^2)A)$.
  - $d(\mathbb{Y}^1) \mathbb{Y}^1$: $d(A^2)(A \odot A^2)$
  - $\mathbb{Y}^1 d(\mathbb{Y}^1)$: $(A \odot A^2)d(A^2)$

- $\mathbb{Y}^3$:

  - $A^s \mathbb{Y}^2$: $Ad(A^4)A$, $Ad(A^2)A^3$, $A^3 d(A^2)A$, $Ad(A^3)A^2$, $A^2 d(A^3)A$, $A^2 d(A^2)A^2$, $Ad(A^2)A$, $Ad(A^2)^2 A$, $Ad(A^2)Ad(A^2)$, $A(A \odot A^2)A$.
  - $\mathbb{Y}^2 A^s$: $d(A^2) Ad(A^2)A$.
  - $\mathbb{Y}^3 \odot \mathbb{Y}^3$: $A \odot (Ad(A^2)A)$.
  - $d(\mathbb{Y}^2) \mathbb{Y}^2$: $d(Ad(A^2)A)A, d(A(A \odot A^2))A, d((A \odot A^2)A)A$
  - $\mathbb{Y}^2 d(\mathbb{Y}^2)$: $Ad((A \odot A^2)A), Ad(A(A \odot A^2))$

- $\mathbb{Y}^4$:

  - $d(\mathbb{Y}^3) \mathbb{Y}^3$: $d(Ad(A^3)A)A$, $d(Ad(A^2)A)A^2$, $d(Ad(A^2)A^2)A$, $d(A^2 d(A^2)A)A$.
  - $\mathbb{Y}^3 d(\mathbb{Y}^3)$: $Ad(Ad(A^3)A)$, $A^2 d(Ad(A^2)A)$, $Ad(A^2 d(A^2)A)$, $Ad(Ad(A^2)A^2)$.

- $\mathbb{Y}^5$:

  - $A^{s_1} d(\mathbb{Y}^3) A^{s_2}$: $Ad(Ad(A^2)A)A$

5-layer PST can express it. $\qquad\square$

Count cycle is closely related to counting path. A $L+1$ cycle contains edge $(i,j)$ can be decomposed into a $L$-path from $i$ to $j$ and edge $(i,j)$. Therefore, the vector of count of cycles rooted in each node $C_{L+1} = \text{diagonal}(AP_L)$

**Theorem 11.** *The diagonal elements of attention matrix of 2-layer PST can express $C_3$, 3-layer PST can express $C_4$, 4-layer PST can express $C_5, C_6$, 6-layer PST can express $C_7$.*

*Proof.* It is a direct conjecture of Theorem 10 as $C_{L+1} = \text{diagonl}(AP_L)$ and $\forall k, P_L \in \mathbb{Y}^k \Rightarrow AP_L \in \mathbb{Y}^{k+1}$ $\qquad\square$

## B    EXPRESSIVITY COMPARISION WITH OTHER MODELS

Algorithm A is considered more expressive than algorithm B if it can differentiate between all pairs of graphs that algorithm B can distinguish. If there is a pair of links that algorithm A can distinguish while B cannot and A is more expressive than B, we say that A is strictly more expressive than B. We will first demonstrate the greater expressiveness of our model by using PST to simulate other models. Subsequently, we will establish the strictness of our model by providing a concrete example.

Our transformer incorporates inner products of coordinates, which naturally allows us to express shortest path distances and various node-to-node distance metrics. These concepts are discussed in more detail in Section 5.1. This naturally leads to the following theorem, which compares our PST with GIN (Xu et al., 2019a).

**Theorem 12.** *A $k$-layer Point Set Transformer is strictly more expressive than a $k$-layer GIN.*

*Proof.* We first prove that one PST layer can simulate an GIN layer.

Given node features $s_i$ and $v_i$. Without loss of generality, we can assume that one channel of $v_i$ contains $U\mathrm{diag}(\Lambda^1/2)$. The sv-mixer can simulate an MLP function applied to $s_i$. Leading to $s_i'$. A GIN layer will then update node representations as follows,

$$s_i \leftarrow s_i' + \sum_{j \in N(i)} s_j' \tag{91}$$

By inner products of coordinates, the attention matrix can express the adjacency matrix. By setting $W_q^s, W_k^s = 0$, and $W_q^v, W_k^v$ be a diagonal matrix with only the diagonal elements at the row corresponding the the channel of $U\mathrm{diag}(\Lambda^1/2)$.

$$K_{ij} = (W_q^s s_i \odot W_k^s s_j) \| \mathrm{diagonal}(W_q^v v_i v_j^T W_k^v) \rightarrow \langle \mathrm{diag}(\Lambda^1/2)U_i, \mathrm{diag}(\Lambda^1/2)U_j \rangle = A_{ij} \tag{92}$$

Let MLP express an identity function.

$$\mathrm{Atten}_{ij} = \mathrm{MLP}(K_{ij}) \rightarrow A_{ij} \tag{93}$$

The attention layer will produce

$$s_i \leftarrow \sum_j A_{ij} s_j' = \sum_{j \in N(i)} s_j' \tag{94}$$

with residual connection, the layer can express GIN

$$s_i \leftarrow s_i' + s_i = s_i' + \sum_{j \in N(i)} s_j' \tag{95}$$

Moreover, as shown in Theorem 5, MPNN cannot compute shortest path distance, while PST can. So PST is strictly more expressive. □

Moreover, our transformer is strictly more expressive than some representative graph transformers, including Graphormer (Ying et al., 2021) and GPS with RWSE as structural encoding (Rampásek et al., 2022).

**Theorem 13.** *A $k$-layer Point Set Transformer is strictly more expressive than a $k$-layer Graphormer and a $k$-layer GPS.*

*Proof.* We first prove that $k$-layer Point Set Transformer is more expressive than a $k$-layer Graphormer and a $k$-layer GPS.

In initialization, besides the original node feature, Graphormer further add node degree features and GPS further utilize RWSE. Our PST can add these features with the first sv-mixer layer.

$$s_i' \leftarrow \mathrm{MLP}_1(s_i \| \mathrm{diagonal}(W_1 v_i v_i^T W_2^T)) \tag{96}$$

Here, $\mathrm{diagonal}(W_1 v_i v_i^T W_2^T)$ add coordinate inner products, which can express RWSE (diagonal elements of random walk matrix) and degree (see Appendix D), to node feature.

Then we are going to prove that one PST layer can express one GPS and one Graphormer layer. PST's attention matrix is as follows,

$$\mathrm{Atten}_{ij} = \mathrm{MLP}(K_{ij}), \quad K_{ij} = (W_q^s s_i \odot W_k^s s_j) \| \mathrm{diagonal}(W_q^v v_i v_j^T W_k^v) \rightarrow \langle \mathrm{diag}(\Lambda^1/2)U_i, \mathrm{diag}(\Lambda^1/2)U_j \rangle \tag{97}$$

The Hadamard product $(W_q^s s_i \odot W_k^s s_j)$ with MLP can express the inner product of node representations used in Graphormer and GPS. The inner product of coordinates can express adjacency matrix used in GPS and Graphormer and shortest path distance used in Graphormer. Therefore, PST' attention matrix can express the attention matrix in GPS and Graphormer.

To prove strictness, Figure 2(c) in (Zhang et al., 2023b) provides an example. As PST can capture resistance distance and simulate 1-WL, so it can differentiate the two graphs according to Theorem

4.2 in (Zhang et al., 2023b). However, Graphormer cannot distinguish the two graphs, as proved in (Zhang et al., 2023b).

For GPS, Two graphs in Figure 2(c) have the same RWSE: RWSE is

$$\text{diagonal}(\hat{U}\text{diag}(\hat{\Lambda}^k)\hat{U}^T), k = 1, 2, 3, ..., \tag{98}$$

where the eigendecomposition of normalized adjacency matrix $D^{-1/2}AD^{-1/2}$ is $\hat{U}$. By computation, we find that two graphs share the same $\hat{\Lambda}$. Moroever, $\text{diagonal}(\hat{U}\text{diag}(\hat{\Lambda}^k)\hat{U}^T)$ are equal in two graphs for $k = 0, 1, 2, ..., 9$, where 9 is the number of nodes in graphs. $\Lambda^k$ and $\text{diagonal}(\hat{U}\text{diag}(\hat{\Lambda}^k)\hat{U}^T)$ with larger $k$ are only linear combinations of $\Lambda^k$ and thus $\text{diagonal}(\hat{U}\text{diag}(\hat{\Lambda}^k)\hat{U}^T)$ for $k = 0, 1, ..., 9$. So the RWSE in the two graphs are equal and equivalent to simply assigning feature $h_1$ to the center node and feature $h_2$ to other nodes in two graphs. Then GPS simply run a model be a submodule of Graphormer on the graph and thus cannot differentiate the two graphs either. □

Even against a highly expressive method such as 2-FWL, our models can surpass it in expressivity with a limited number of layers:

**Theorem 14.** *For all $K > 0$, a graph exists that a $K$-iteration 2-FWL method fails to distinguish, while a 1-layer Point Set Transformer can.*

*Proof.* It is a direct corollary of Theorem 4. □

## C  SOME GRAPH TRANSFORMERS FAIL TO COMPUTE SHORTEST PATH DISTANCE

First, we demonstrate that computing inner products of node representations alone cannot accurately determine the shortest path distance when the node representations are permutation-equivariant. Consider Figure 2 as an illustration. In cases where node representations exhibit permutation-equivariance, nodes $v_2$ and $v_3$ will share identical representations. Consequently, the pairs $(v_1, v_2)$ and $(v_1, v_3)$ will yield the same inner products of node representations, despite having different shortest path distances. Consequently, it becomes evident that the attention matrices of some Graph Transformers are incapable of accurately computing the shortest path distance.

**Theorem 15.** *GPS with RWSE (Rampásek et al., 2022) and Graphormer without shortest path distance encoding cannot compute shortest path distance with the elements of adjacency matrix.*

*Proof.* Their adjacency matrix elements are functions of the inner products of node representations and the adjacency matrix.

$$\text{Atten}_{ij} = \langle s_i, s_j \rangle \| A_{ij}. \tag{99}$$

This element is equal for the node pair $(v_1, v_2)$ and $(v_1, v_3)$ in Figure 2. However, two pairs have different shortest path distances. □

Furthermore, while Graph Transformers gather information from the entire graph, they may not have the capacity to emulate multiple MPNNs with just a single transformer layer. To address this, we introduce the concept of a *vanilla Graph Transformer*, which applies a standard Transformer to nodes using the adjacency matrix for relative positional encoding. This leads us to the following theorem.

**Theorem 16.** *For all $k \in \mathbb{N}$, there exists a pair of graph that $k + 1$-layer MPNN can differentiate while $k$-layer MPNN and $k$-layer vanilla Graph Transformer cannot.*

*Proof.* Let $H_l$ denote a circle of $l$ nodes. Let $G_l$ denote a graph of two components, one is $H_{\lfloor l/2 \rfloor}$ and the other is $H_{\lceil l/2 \rceil}$. Let $H_l'$ denote adding a unique feature 1 to a node in $H_l$ (as all nodes are symmetric for even $l$, the selection of node does not matter), and $G_l'$ denote adding a unique feature 1 to one node in $G_l$. All other nodes have feature 0. Now we prove that

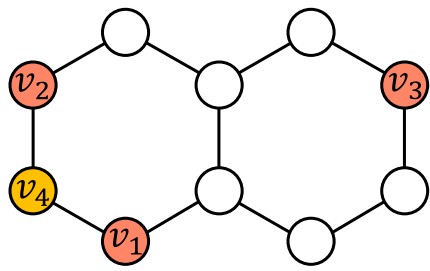

Figure 2: The failure of using inner products of permutation-equivariant node representations to predict shortest path distance. $v_2$ and $v_3$ have equal node representations due to symmetry. Therefore, $(v_1, v_2)$ and $(v_1, v_3)$ will have the same inner products of node representations but different shortest path distance.

**Lemma 10.** *For all $K \in \mathbb{N}$, $(K+1)$-layer MPNN can distinguish $H'_{4(K+1)}$ and $G'_{4(K+1)}$, while $K$-layer MPNN and $K$-layer vanilla Graph Transformer cannot distinguish.*

Given $H'_{4(K+1)}$, $G'_{4(K+1)}$, we assign each node a color according to its distance to the node with extra label 1: $c_0$ (the labeled node itself), $c_1$ (two nodes connected to the labeled node), $c_2$ (two nodes whose shortest path distance to the labeled node is 2),..., $c_K$ (two nodes whose shortest path distance to the labeled node is $K$), $c_{>K}$ (nodes whose shortest path distance to the labeled node is larger than $K$). Now by simulating the process of MPNN, we prove that at $k$-th layer $k <= K$, $\forall i \leq k$, $c_i$ nodes have the same representation (denoted as $h_i^k$), respectively, $c_{k+1}, c_{k+2}, ..., c_K, c_{>K}$ nodes all have the same representation (denoted as $h_{k+1}^k$).

Initially, all $c_0$ nodes have representation $h_0^0$, all other nodes have representation $h_1^0$ in both graph.

Assume at $k$-th layer, $\forall i \leq k$, $c_i$ nodes have the same representation $h_i^k$, respectively, $c_{k+1}, c_{k+1}, ..., c_K, c_{>K}$ nodes all have the same representation $h_{k+1}^k$. At $k + 1$-th layer, $c_0$ node have two neighbors of representation $h_1^k$. all $c_i, 1 < i <= k$ node two neighbors of representations $h_{i-1}^k$ and $h_{i+1}^k$, respectively. $c_{k+1}$ nodes have two neighbors of representation $h_k^k$ and $h_{k+1}^k$. All other nodes have two neighbors of representation $h_{k+1}^k$. So $c_i, i \leq k + 1$ nodes have the same representation (denoted as $h_i^{k+1}$), respectively, $c_{k+1+1}, ..., c_K, c_{>K}$ nodes all have the same representation (denoted as $h_{k+1}^k$).

The same induction also holds for $K$-layer vanilla graph transformer.

However, in the $K + 1$-th message passing layer, only one node in $G_{4(K+1)}$ is of shortest path distance $K + 1$ to the labeled node. It also have two neighbors of representation $h_K^K$. While such a node is not exist in $H_{4(K+1)}$. So $(K + 1)$-layer MPNN can distinguish them. □

The issue with a vanilla Graph Transformer is that, although it collects information from all nodes in the graph, it can only determine the presence of features in 1-hop neighbors. It lacks the ability to recognize features in higher-order neighbors, such as those in 2-hop or 3-hop neighbors. A straightforward solution to this problem is to manually include the shortest path distance as a feature. However, our analysis highlights that aggregating information from the entire graph is insufficient for capturing long-range interactions.

# D  CONNECTION WITH STRUCTURAL EMBEDDINGS

We show the equivalence between the structural embeddings and the inner products of our generalized coordinates in Table 5.

# E  DATASETS

We summarize the statistics of our datasets in Table 6. Synthetic is the dataset used in substructure counting tasks provided by Huang et al. (2023), they are random graph with the count of substructure as node label. QM9 (Wu et al., 2017), ZINC (Gómez-Bombarelli et al., 2016), and ogbg-molhiv are three datasets of small molecules. QM9 use 13 quantum chemistry property as the graph label.

Table 5: Connection between existing structural embeddings and our parameterized coordinates. The eigendecomposition are $\hat{A} \leftarrow \hat{U}\hat{\Lambda}\hat{U}$, $D - A \leftarrow \tilde{U}\tilde{\Lambda}\tilde{U}^T$, $A \leftarrow U\Lambda U^T$. $d_i$ denote the degree of node $i$.

| Method | Description | Connection |
|---|---|---|
| Random walk matrix (Li et al., 2020; Dwivedi et al., 2023; Rampásek et al., 2022) | $k$-step random walk matrix is $(D^{-1}A)^k$, whose element $(i,j)$ is the probability that a $k$-step random walk starting from node $i$ ends at node $j$. | $(D^{-1}A)^k_{ij}$ $= (D^{-1/2}(\hat{A})^k D^{1/2})_{ij}$ $= \sqrt{\frac{d_j}{d_i}}\langle\hat{U}_i, \mathrm{diag}(\hat{\Lambda}^k)\hat{U}_j\rangle$ |
| Heat kernel matrix (Mialon et al., 2021) | Heat kernel is a solution of the heat equation. Its element $(i,j)$ represent how much heat diffuse from node $i$ to node $j$ | $(I+\tilde{U}(\mathrm{diag}(\exp(-t\tilde{\Lambda}))-I)\tilde{U}^T)_{ij}$ $= \delta_{ij} + \langle\tilde{U}_i, (\mathrm{diag}(\exp(-t\tilde{\Lambda})) - I)\tilde{U}_j\rangle$ |
| Resistance distance (Zhang & Li, 2021; Zhang et al., 2023b) | Its element $(i,j)$ is the resistance between node $i,j$ considering the graph as an electrical network. It is also the pseudo-inverse of laplacian matrix $L$, | $(\tilde{U}\mathrm{diag}(\tilde{\Lambda}^{-1})\tilde{U}^T)_{ij}$ $= \langle\tilde{U}_i, \mathrm{diag}(\tilde{\Lambda}^{-1})\tilde{U}_j\rangle$ |
| Equivariant and stable laplacian PE (Wang et al., 2022) | The encoding of node pair $i,j$ is $\|1_K \odot (U_i - U_j)\|$, where $1_K$ means a vector $\in \mathbb{R}^r$ with its elements coresponding to $K$ largest eigenvalue of $L$ | $\|1_K \odot (U_i - U_j)\|^2$ $= \langle U_i, \mathrm{diag}(1_K)U_i\rangle$ $+ \langle U_j, \mathrm{diag}(1_K)U_j\rangle$ $- 2\langle U_i, \mathrm{diag}(1_K)U_j\rangle$ |
| Degree and number of triangular (Bouritsas et al., 2023) | $d_i$ is the number of edges, and $t_i$ is the number of triangular rooted in node $i$. | $d_i = \langle U_i, \mathrm{diag}(\Lambda^2)U_j\rangle$. $t_i = \langle U_i, \mathrm{diag}(\Lambda^3)U_j\rangle$ |

Table 6: Statistics of the datasets. #Nodes and #Edges denote the number of nodes and edges per graph. In split column, 'fixed' means the dataset uses the split provided in the original release. Otherwise, it is of the formal training set ratio/valid ratio/test ratio.

| | #Graphs | #Nodes | #Edges | Task | Metric | Split |
|---|---|---|---|---|---|---|
| Synthetic | 5,000 | 18.8 | 31.3 | Node Regression | MAE | 0.3/0.2/0.5. |
| QM9 | 130,831 | 18.0 | 18.7 | Regression | MAE | 0.8/0.1/0.1 |
| ZINC | 12,000 | 23.2 | 24.9 | Regression | MAE | fixed |
| ZINC-full | 249,456 | 23.2 | 24.9 | Regression | MAE | fixed |
| ogbg-molhiv | 41,127 | 25.5 | 27.5 | Binary classification | AUC | fixed |
| PascalVOC-SP | 11,355 | 479.4 | 2710.5 | Node Classification | Macro F1 | fixed |
| Peptides-func | 15,535 | 150.9 | 307.3 | Classification | AP | fixed |
| Peptides-struct 1 | 15,535 | 150.9 | 307.3 | Regression | MAE | fixed |

It provides both the graph and the coordinates of each atom. ZINC provides graph structure only and aim to predict constrained solubility. Ogbg-molhiv is one of Open Graph Benchmark dataset, which aims to use graph structure to predict whether a molecule can inhibits HIV virus replication. Besides these datasets of small molecules, we also use three datasets in Long Range Graph Benchmark (Dwivedi et al., 2022b). They consists of larger graphs. PascalVOC-SP comes from the computer vision domain. Each node in it representation a superpixel and the task is to predict the semantic segmentation label for each node. Peptide-func and peptide struct are peptide molecular graphs. Task in Peptides-func is to predict the peptide function. Peptides-struct is to predict 3D properties of the peptide.

## F EXPERIMENTAL DETAILS

Our code is primarily based on PyTorch (Paszke et al., 2019) and PyG (Fey & Lenssen, 2019). All our experiments are conducted on NVIDIA RTX 3090 GPUs on a linux server. We select the

Table 7: Hyperparameters of our model for each dataset. #warm means the number of warmup epochs, #cos denotes the number of cosine annealing epochs, gn denotes the magnitude of the gaussian noise injected into the point coordinates, hiddim denotes hidden dimension, bs means batch size, lr represents learning rate, and #epoch is the number of epochs for training.

| dataset | #warm | #cos | wd | gn | #layer | hiddim | bs | lr | #epoch | #param |
|---|---|---|---|---|---|---|---|---|---|---|
| Synthetic | 10 | 15 | 6e-4 | 1e-6 | 9 | 96 | 16 | 0.0006 | 300 | 961k |
| qm9 | 1 | 40 | 1e-1 | 1e-5 | 8 | 128 | 256 | 0.001 | 150 | 1587k |
| ZINC | 17 | 17 | 1e-1 | 1e-4 | 6 | 80 | 128 | 0.001 | 800 | 472k |
| ZINC-full | 40 | 40 | 1e-1 | 1e-6 | 8 | 80 | 512 | 0.003 | 400 | 582k |
| ogbg-molhiv | 5 | 5 | 1e-1 | 1e-6 | 6 | 96 | 24 | 0.001 | 300 | 751k |
| PascalVOC-SP | 5 | 5 | 1e-1 | 1e-5 | 4 | 96 | 6 | 0.001 | 40 | 527k |
| Peptide-func | 40 | 20 | 1e-1 | 1e-6 | 6 | 128 | 2 | 0.0003 | 80 | 1337k |
| Peptide-struct | 40 | 20 | 1e-1 | 1e-6 | 6 | 128 | 2 | 0.0003 | 40 | 1337k |

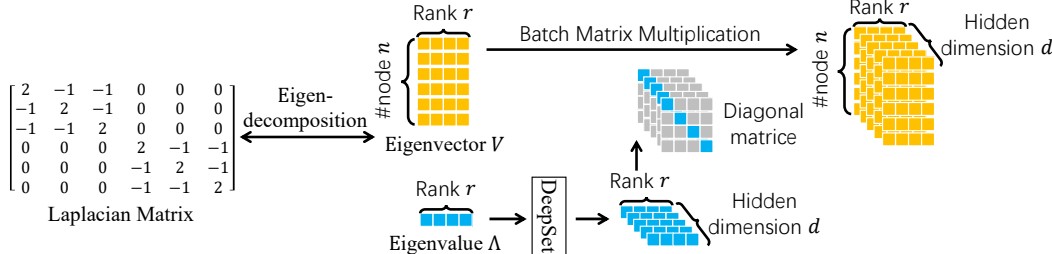

Figure 3: The pipeline of parameterized SRD. We first decompose Laplacian matrix or other matrice for the non-zero eigenvalue and the corresponding eigenvectors. Then the eigenvalue is transformed with DeepSet (Segol & Lipman, 2020). Multiply the transformed eigenvalue and the eigenvector leads to coordinates.

hyperparameters by running optuna (Akiba et al., 2019) to optimize the validation score. We run each experiment with 8 different seeds, reporting the averaged results at the epoch achieving the best validation metric. For optimization, we use AdamW optimizer and cosine annealing scheduler. Hyperparameters for datasets are shown in Table 7. All PST models (except these in ablation study) decompose laplacian matrix for coordinates.

**ZINC, ZINC-full, PascalVOC-SP, Peptide-func, and Peptide-struct have 500k parameter budgets. Other datasets have no parameter limit. Graphormer (Ying et al., 2021) takes 47000k parameters on ogbg-molhiv. 1-2-3-GNN takes 929k parameters on qm9.**

## G ARCHITECTURE

The architecture of parameterized SRD is shown in Figure **??**. The architecture of PST is shown in Figure 4.

## H ABLATION

To assess the design choices made in our Point Set Transformer (PST), we conducted ablation experiments. First, we removed the generalized coordinates (see Section 3.2) from our model, resulting in a reduced version referred to as the PST-gc model. Additionally, we introduced a variant called PST-onelayer, which is distinct from PST in that it only computes the attention matrix once and does not combine information in scalar and vector features. Furthermore, PST decompose Laplacian matrix by default to produce coordinates. PST-adj uses adjacency matrix instead. Similar to PST, DeepSet takes node coordinates as input. However, it use DeepSet (Segol & Lipman, 2020)

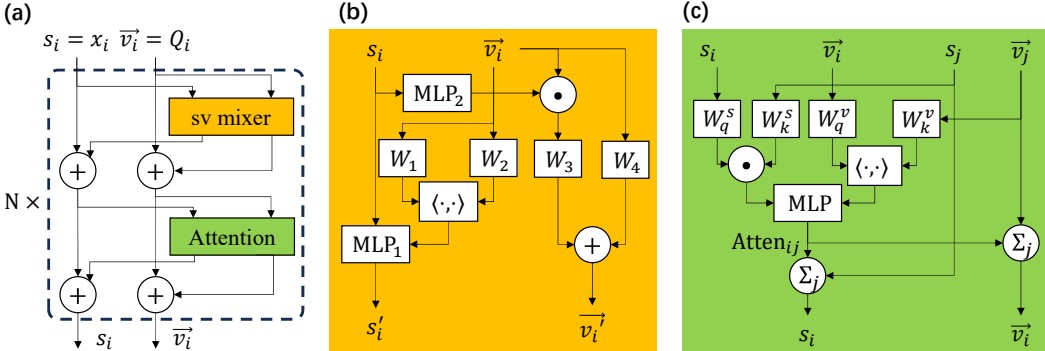

Figure 4: Architecture of Point Set Transformer (PST) (a) PST contains several layers. Each layer is composed of an scalar-vector (sv)-mixer and an attention layer. (b) The architecture of sv-mixer. (c) The architecture of attention layer. $s_i$ and $s_i'$ denote the scalar representations of node $i$, and $\vec{v}_i$ and $\vec{v}_i'$ denote the vector representations. $x_i$ is the initial features of node $i$. $Q_i$ and point coordinates of node $i$ produced by parameterized SRD in Section 3.2.

Table 8: Ablation study on qm9 dataset.

| | $\mu$ | $\alpha$ | $\varepsilon_{\text{homo}}$ | $\varepsilon_{\text{lumo}}$ | $\Delta\varepsilon$ | $R^2$ | ZPVE | $U_0$ | $U$ | $H$ | $G$ | $C_v$ |
|---|---|---|---|---|---|---|---|---|---|---|---|---|
| Unit | $10^{-1}$D | $10^{-1}a_0^3$ | $10^{-2}$meV | $10^{-2}$meV | $10^{-2}$meV | $a_0^2$ | $10^{-2}$meV | meV | meV | meV | meV | $10^{-2}$cal/mol/K |
| PST | $3.19_{\pm 0.04}$ | $1.89_{\pm 0.04}$ | $5.98_{\pm 0.09}$ | $5.84_{\pm 0.08}$ | $8.46_{\pm 0.07}$ | $13.08_{\pm 0.16}$ | $0.39_{\pm 0.01}$ | $3.46_{\pm 0.17}$ | $3.55_{\pm 0.10}$ | $3.49_{\pm 0.20}$ | $3.55_{\pm 0.17}$ | $7.77_{\pm 0.15}$ |
| PST-onelayer | $3.72_{\pm 0.02}$ | $2.25_{\pm 0.05}$ | $6.62_{\pm 0.11}$ | $6.67_{\pm 0.07}$ | $9.37_{\pm 0.15}$ | $15.95_{\pm 0.29}$ | $0.55_{\pm 0.01}$ | $3.46_{\pm 0.06}$ | $3.50_{\pm 0.14}$ | $3.50_{\pm 0.03}$ | $3.45_{\pm 0.07}$ | $9.62_{\pm 0.24}$ |
| PST-gc | $3.34_{\pm 0.02}$ | $1.93_{\pm 0.06}$ | $6.08_{\pm 0.11}$ | $6.10_{\pm 0.10}$ | $8.65_{\pm 0.10}$ | $13.71_{\pm 0.12}$ | $0.40_{\pm 0.01}$ | $3.38_{\pm 0.13}$ | $3.43_{\pm 0.12}$ | $3.33_{\pm 0.08}$ | $3.29_{\pm 0.11}$ | $8.04_{\pm 0.15}$ |
| PST-adj | $3.16_{\pm 0.02}$ | $1.86_{\pm 0.01}$ | $6.31_{\pm 0.06}$ | $6.10_{\pm 0.05}$ | $8.84_{\pm 0.01}$ | $13.60_{\pm 0.09}$ | $0.39_{\pm 0.01}$ | $3.59_{\pm 0.12}$ | $3.73_{\pm 0.08}$ | $3.65_{\pm 0.06}$ | $3.60_{\pm 0.016}$ | $7.62_{\pm 0.21}$ |
| PST-normadj | $3.22_{\pm 0.04}$ | $1.85_{\pm 0.02}$ | $5.97_{\pm 0.23}$ | $6.15_{\pm 0.07}$ | $8.79_{\pm 0.04}$ | $13.42_{\pm 0.15}$ | $0.41_{\pm 0.01}$ | $3.36_{\pm 0.25}$ | $3.41_{\pm 0.24}$ | $3.46_{\pm 0.18}$ | $3.38_{\pm 0.23}$ | $8.10_{\pm 0.12}$ |
| DeepSet | $3.53_{\pm 0.05}$ | $2.05_{\pm 0.02}$ | $6.56_{\pm 0.03}$ | $6.31_{\pm 0.05}$ | $9.13_{\pm 0.04}$ | $14.35_{\pm 0.02}$ | $0.41_{\pm 0.02}$ | $3.53_{\pm 0.11}$ | $3.49_{\pm 0.05}$ | $3.47_{\pm 0.04}$ | $3.56_{\pm 0.14}$ | $8.35_{\pm 0.09}$ |
| DF | $3.46$ | $2.22$ | $6.15$ | $6.12$ | $8.82$ | $15.04$ | $0.46$ | $4.24$ | $4.16$ | $3.95$ | $4.24$ | $9.01$ |

rather than transformer as the set encoder. For better comparison, we also use our strongest baseline on QM9 dataset, DF (Zhou et al., 2023).

The results of the ablation study conducted on the QM9 dataset are summarized in Table 2. Notably, PST-gc exhibits only a slight increase in test loss compared to PST, and even outperforms PST on 4 out of 12 target metrics, highlighting the effectiveness of the Graph as Point Set approach with vanilla symmetric rank decomposition. In contrast, PST-onelayer performs significantly worse, underscoring the advantages of PST over previous methods that augment adjacency matrices with spectral features. PST-adj and PST-normadj achieves similar performance to PST, illustrating that the choice of matrix to decompose does not matter. DeepSet performs worse than PST, but it still outperforms our strongest baseline DF, showing the potential of combining set encoders other than transformer with our convertion from graph to set. On the long-range graph benchmark, PST maintains a significant performance edge over PST-onelayer. However, it's worth noting that the gap between PST and PST-gc widens, further confirming the effectiveness of gc in modeling long-range interactions.

## I    SCALABILITY

We present training time per epoch and GPU memory consumption data in Table 10.On the ZINC dataset, PST ranks as the second fastest model, and its memory consumption is comparable to existing models with strong expressivity, such as SUN and SSWL, and notably lower than PPGN.

## J    RESULTS ON TU DATASETS

Following the setting of Feng et al. (2022), we test our PST on four TU datasets (Ivanov et al., 2019). The results are shown in Table 12. Baselines include WL subtree kernel (Shervashidze et al., 2011), GIN (Xu et al., 2019a), DGCNN (Zhang et al., 2018), GraphSNN (Wijesinghe & Wang, 2022), GNN-AK+ (Zhao et al., 2022), and three variants of KP-GNN (Feng et al., 2022) (KP-GCN,

Table 9: Ablation study on Long Range Graph Benchmark dataset.

| Model | PascalVOC-SP | Peptides-Func | Peptides-Struct |
|---|---|---|---|
| PST | $0.4010_{\pm 0.0072}$ | $0.6984_{\pm 0.0051}$ | $0.2470_{\pm 0.0015}$ |
| PST-onelayer | $0.3229_{\pm 0.0051}$ | $0.6517_{\pm 0.0076}$ | $0.2634_{\pm 0.0019}$ |
| PST-gc | $0.4007_{\pm 0.0039}$ | $0.6439_{\pm 0.0342}$ | $0.2564_{\pm 0.0120}$ |

Table 10: Training time per epoch and GPU memory consumption on zinc dataset with batch size 128.

| | PST | SUN | SSWL | PPGN | Graphormer | GPS | SAN-GPS |
|---|---|---|---|---|---|---|---|
| Time/s | 15.20 | 20.93 | 45.30 | 20.21 | 123.79 | 11.70 | 79.08 |
| Memory/GB | 4.08 | 3.72 | 3.89 | 20.37 | 0.07 | 0.25 | 2.00 |

KP-GraphSAGE, and KP-GIN). We use 10-fold cross-validation, where 9 folds are for training and 1 fold is for testing. The average test accuracy is reported. Our PST consistently outperforms our baselines.

Table 11: Training time per epoch and GPU memory consumption on pascalvoc-sp dataset with batch size 6.

|           | PST   | SUN   | SSWL  | PPGN  | Graphormer | GPS   | SAN-GPS |
|-----------|-------|-------|-------|-------|------------|-------|---------|
| Time/s    | 15.20 | 20.93 | 45.30 | 20.21 | 123.79     | 11.70 | 79.08   |
| Memory/GB | 4.08  | 3.72  | 3.89  | 20.37 | 0.07       | 0.25  | 2.00    |

Table 12: TU dataset evaluation result.

| Method       | MUTAG        | PTC-MR       | PROTEINS      | IMDB-B       |
|--------------|--------------|--------------|---------------|--------------|
| **WL**       | 90.4±5.7     | 59.9±4.3     | 75.0±3.1      | 73.8±3.9     |
| **GIN**      | 89.4±5.6     | 64.6±7.0     | 75.9±2.8      | 75.1±5.1     |
| **DGCNN**    | 85.8±1.7     | 58.6 ±2.5    | 75.5±0.9      | 70.0±0.9     |
| **GraphSNN** | 91.24±2.5    | 66.96±3.5    | 76.51±2.5     | 76.93±3.3    |
| **GIN-AK+**  | 91.30±7.0    | 68.20±5.6    | 77.10±5.7     | 75.60±3.7    |
| **KP-GCN**       | 91.7±6.0 | 67.1±6.3     | 75.8±3.5      | 75.9±3.8     |
| **KP-GraphSAGE** | 91.7±6.5 | 66.5±4.0     | 76.5±4.6      | 76.4±2.7     |
| **KP-GIN**       | 92.2±6.5 | 66.8±6.8     | 75.8±4.6      | 76.6±4.2     |
| **PST**      | **94.4±3.5** | **68.8±4.6** | **80.7±3.5**  | **78.9±3.6** |

