# OpenReview forum: "Graph as Point Set"
_ICLR.cc/2024/Conference — Submitted to ICLR 2024_

### Official Review · Reviewer_8zdN · 2023-10-24

**Soundness:** 3 good
**Presentation:** 2 fair
**Contribution:** 3 good
**Rating:** 6
**Confidence:** 5

**Summary:**

This paper presents a method for representing graph data as a set of
points. The representation is obtained based on symmetric rank
decomposition, followed by the application of a permutation-equivariant
function. Subsequently, the resulting features are treated as
a 'coordinate representation' of the graph, and a Transformer model is
used to address downstream tasks.

**Strengths:**

- Finding novel ways to represent graphs, in particular methods that do
  not make use of message passing, are advantageous since they often
  pinpoint novel research directions and permit escaping the WL
  hierarchy for graph expressivity.

- The method is elegant and grounded in strong theory (SRD).

**Weaknesses:**

The current write-up is suffering from substantial flaws, which cannot
be easily rectified during the conference cycle:

1. Lack of clarity: there are some (minor) issues with notation (for
   instance, $Z$ is used both as a matrix and as
   a permutation-equivariant function), but the overall description of
   the method should be clarified. I needed multiple reads to understand
   just *how* a graph is transformed and, moreover, understand the
   algorithmic details of this transformation.

   An overview figure or a brief summary would be immensely helpful
   here.

2. Lack of experimental depth: given the strong claims made in the paper
   concerning a new state of the art, a much more detailed experimental
   setup is required. In particular, a comparison of the different
   models in terms of the number of parameters (of the model itself) and
   the number of hyperparameters is required. Given recent work on
   [improvements in the LRGB data
   set](https://arxiv.org/pdf/2309.00367.pdf), which are only contingent
   on hyperparameter tuning, a more detailed explanation of the setup
   and the comparison is required.

There are also some minor weaknesses:

- Table 1 is hard to read and understand at first glance. Please either
  highlight the best/second-best method or refrain from highlighting
  altogether. Also, why are there no standard deviations for the other
  methods?

- There are some language issues, which would require an additional pass
  over the paper (this is a minor point but it nevertheless slightly
  impacts accessibility).

**Questions:**

1. How does the proposed approach compare to the work by [Kim et
   al.](https://arxiv.org/abs/2110.14416), which also employs
   Transformers for graph learning tasks?

2. What are the contributions of the Transformer architecture to
   predictive performance? Would a simple set function, realised by an
   MLP with an appropriate pooling function, work as well? I believe
   that disentangling and ablating the results would strengthen the
   paper immensely.

3. What is $X$ in Theorem 1 (and the subsequent text)? I assume that
   this pertains to the features of the graph?

4. To what extent is the method contingent on the SRD? Would it be
   possible to use another type of decomposition?

---

> ### Author Response · Authors · 2023-11-19
> **Response to Reviewer 8zdN (1/2)**
>
> We are grateful for Reviewer 8zdN's comprehensive feedback and have addressed their concerns as follows:
>
> 1. **W1-1:** Lack of clarity: there are some (minor) issues with notation (for instance, $Z$ is used both as a matrix and as a permutation-equivariant function),
>
> We appreciate this observation. To resolve this confusion, we have changed the notation from $Z$ to $\mathcal{Z}$ in our revised manuscript.
>
> 2. **W1-2:** but the overall description of the method should be clarified. I needed multiple reads to understand just *how* a graph is transformed and, moreover, understand the algorithmic details of this transformation. An overview figure or a brief summary would be immensely helpful here.
>
> To aid in understanding, Figure 1 provides a comprehensive overview of our method. Additionally, Figure 3 in Appendix G summarizes our Point Set Transformer (PST), which we hope will further clarify our approach.
>
> 3. **W2** Lack of experimental depth: given the strong claims made in the paper concerning a new state of the art, a much more detailed experimental setup is required. In particular, a comparison of the different models in terms of the number of parameters (of the model itself) and the number of hyperparameters is required. Given recent work on improvements in the LRGB data set [1], which are only contingent on hyperparameter tuning, a more detailed explanation of the setup and the comparison is required.
>
> Thank you for the insightful comment. We have updated our manuscript to include more comprehensive experimental details, as you suggested. The updates are reflected in Appendix E, where we provide information about both the number of parameters in the models and the hyperparameters used in our experiments.
>
> 1. Parameter Budget Across Datasets:
>    - For the zinc, zinc-full, and pascalvoc-sp datasets, we adhered to the baseline parameter budget constraint of approximately 500k parameters.
>    - On the peptide-func and peptide-struct datasets, we mistakenly doubled the number of parameters. This occurred because our PST employs two sets of parameters (one for scalar and one for vector), which resulted in twice the parameters with the same hidden dimension and number of transformer layers. We are currently conducting experiments with our PST with smaller hidden dimension and baseline with larger hidden dimensions for these datasets. The results are as followings. Compared with representative graph transformers, our PST performs worse than baselines with parameter 500 K. However, when the PST is scaled to 1M parameters, its performance becomes comparable to, or even surpasses, that of other baseline models like Graph-MLPMixer and GraphGPS. Therefore, our PST can still outperforms baselines with the same parameter budget 1M, and our method is effective on the two datasets.
> 	| | peptide-func | peptide-struct |
> 	| --- | --- | --- |
> 	| \#param=500K | | |
> 	| Graph-MLPMixer | $0.6970_{\pm0.0080}$ | $0.2475_{\pm0.0015}$ |
> 	| GraphGPS | $0.6535_{\pm0.0041}$ | $0.2500_{\pm0.0005}$ |
> 	| PST | $0.6344_{\pm0.0041}$ | $0.2543_{\pm 0.0014}$ |
> 	| \#param=1M | | |
> 	| Graph-MLPMixer | $0.7004_{\pm 0.0040}$ | $0.2503_{\pm 0.0031}$ |
> 	| GraphGPS | $0.6561_{\pm 0.0027}$ | $0.2557_{\pm 0.0010}$ |
> 	| PST | $0.6984_{\pm0.0051}$ | $\mathbf{0.2470_{\pm 0.0015}}$ |
>    - Other datasets we explored do not have explicit parameter constraints, and it's worth noting that our PST has fewer parameters compared to representative baselines in these cases.
>
> 2. Hyperparameter Configuration:
>    - Our experiments involved tuning seven hyperparameters: depth (4, 8), hidden dimension (64, 256), learning rate (0.0001, 0.003), number of warmup epochs (0, 40), number of cosine annealing epochs (1, 20), magnitude of Gaussian noise added to input (1e-6, 1e-4), and weight decay (1e-6, 1e-1).
>    - We observed that [1] also used seven hyperparameters in their setup.
>    - Batch size was determined based on GPU memory constraints and was not a parameter that we optimized.
>
> [1] Jan Tönshoff et al. Where Did the Gap Go? Reassessing the Long-Range Graph Benchmark. https://arxiv.org/pdf/2309.00367.pdf
>
>
> 4. **W3** Table 1 is hard to read and understand at first glance. Please either highlight the best/second-best method or refrain from highlighting altogether. Also, why are there no standard deviations for the other methods?
>
> In the revised manuscript, we have clarified that highlighting in Table 1 indicates models capable of counting the corresponding substructure (test loss < 10 units). This is now explicitly stated for better understanding. As for the lack of standard deviations in other methods, those results are directly sourced from their original papers, which did not provide standard deviations.

---

> > ### Author Response · Authors · 2023-11-19
> > **Response to Reviewer 8zdN (2/2)**
> >
> > 5. **W4** There are some language issues, which would require an additional pass over the paper (this is a minor point but it nevertheless slightly impacts accessibility).
> >
> > We have addressed the issues identified by the reviewers and will continue to refine the language for greater clarity and readability.
> >
> > 6. **Q1** How does the proposed approach compare to the work by [Kim et al.](https://arxiv.org/abs/2110.14416), which also employs Transformers for graph learning tasks?
> >
> > Our methodology differs significantly from Kim et al. and other existing graph transformers, which focus on encoding adjacency matrices. They still take adjacency matrices (or variants like random walk matrices) as input. In contrast, our PST employs a set model that operates without adjacency input, encoding a set of independent points with coordinates. When tested on the zinc dataset using the official code and settings of Kim et al., our PST significantly outperforms their early model.
> >
> > | Model | test l1 loss | training time/epoch | GPU memory |
> > | ---- | ---- | ---- | ---- |
> > | [Kim et al.](https://arxiv.org/abs/2110.14416) | 0.150±0.004  | 148 s | 9.24 G     |
> > | PST | 0.063±0.003  | 15 s | 4.08 G |
> >
> > 7. **Q2:** What are the contributions of the Transformer architecture to predictive performance? Would a simple set function, realised by an MLP with an appropriate pooling function, work as well? **I believe that disentangling and ablating the results would strengthen the paper immensely.**
> >
> > We have conducted an ablation study on the role of the Transformer in our model, detailed in Appendix H. The study compares our PST with a DeepSet encoder and our strongest baseline DF on the QM9 dataset. The results show that while the Transformer in PST offers superior performance, the DeepSet encoder still outperforms the baseline on 8/12 targets, demonstrating the effectiveness of our Graph As Set approach.
> >
> > | ~ | $\mu$ | $\alpha$ | $\varepsilon_{\text{homo}}$ | $\varepsilon_{\text{lumo}}$ | $\Delta\varepsilon$ | $R^2$ | ZPVE | $U_0$ | $U$ | $H$ | $G$ | $C_v$ |
> > |---|---|---|---|---|---|---|---|---|---|---|---|---|
> > | Unit | $10^{-1}$D | $10^{-1}$$a_0^3$ | $10^{-2}$meV | $10^{-2}$meV | $10^{-2}$meV | $a_0^2$ | $10^{-2}$meV | meV | meV | meV | meV | $10^{-2}$cal/mol/K |
> > | PST | $3.19_{\pm 0.04}$ | $1.89_{\pm 0.04}$ | $5.98_{\pm 0.09}$ | $5.84_{\pm 0.08}$ | $8.46_{\pm 0.07}$ | $13.08_{\pm 0.16}$ | $0.39_{\pm 0.01}$ | $3.46_{\pm 0.17}$ | $3.55_{\pm 0.10}$ | $3.49_{\pm 0.20}$ | $3.55_{\pm 0.17}$ | $7.77_{\pm 0.15}$ |
> > | DeepSet | $3.53_{\pm 0.05}$ | $2.05_{\pm 0.02}$ | $6.56_{\pm 0.03}$ | $6.31_{\pm 0.05}$ | $9.13_{\pm 0.04}$ | $14.35_{\pm 0.02}$ | $0.41_{\pm 0.02}$ | $3.53_{\pm 0.11}$ | $3.49_{\pm 0.05}$ | $3.47_{\pm 0.04}$ | $3.56_{\pm 0.14}$ | $8.35_{\pm 0.09}$ |
> > | DF | $3.46$ | $2.22$ | $6.15$ | $6.12$ | $8.82$ | $15.04$ | $0.46$ | $4.24$ | $4.16$ | $3.95$ | $4.24$ | $9.01$ |
> >
> >
> > 8. **Q3:** What is $X$ in Theorem 1 (and the subsequent text)? I assume that this pertains to the features of the graph?
> >
> > Yes, $X$ refers to the node feature matrix of the graph, as defined in the preliminary section of our submission.
> >
> > 9. **Q4:** To what extent is the method contingent on the SRD? Would it be possible to use another type of decomposition?
> >
> > Our method hinges on the properties outlined in Proposition 1, which allows for the complete conversion of graph problems into set problems. To our knowledge, SRD and its parameterized versions are the only decompositions that fulfill these criteria. For example, eigendecomposition does not offer the same level of efficacy.

---

> > > ### Comment · Reviewer_8zdN · 2023-11-22
> > >
> > > I thank the authors for the extensive rebuttal and apologise for the delay—a colleague passed away unexpectedly and things have been hectic.
> > >
> > > In any case, I appreciate the detailed responses and, conditional on further improvements to clarity (some restructuring so that the most relevant details are in the main paper), I will raise my score accordingly. I believe this has the potential to be a strong contribution to the literature.

---

> > > > ### Author Response · Authors · 2023-11-22
> > > >
> > > > Thank you for your response. I'm truly sorry to hear about your colleague's passing, and I understand that this has been a difficult time for you. Your feedback is invaluable, and we appreciate your understanding and patience.
> > > >
> > > > We have made the following updates to address your concerns:
> > > >
> > > > 1. $Z$ is used both as a matrix and as a permutation-equivariant function.
> > > >
> > > > We have changed the notation from $Z$ to $\mathcal{Z}$ in Theorem 1 to avoid confusion, where $Z$ is used both as a matrix and as a permutation-equivariant function.
> > > >
> > > > 2.  I needed multiple reads to understand just how a graph is transformed and, moreover, understand the algorithmic details of this transformation. An overview figure or a brief summary would be immensely helpful here.
> > > >
> > > > To provide a clearer understanding of how a graph is transformed and the algorithmic details, we have added a figure to Figure 3 in Appendix G. Additionally, we've included a summary in the end of Section 3 and referred to the figure to aid in comprehension.
> > > >
> > > > 3. a much more detailed experimental setup is required. In particular, a comparison of the different models in terms of the number of parameters (of the model itself) and the number of hyperparameters is required. Given recent work on improvements in the LRGB data set, which are only contingent on hyperparameter tuning, a more detailed explanation of the setup and the comparison is required.
> > > >
> > > > In response to your request for a more detailed experimental setup, we have included an illustration of the parameter budget in the caption of the table for the ZINC, ZINC-FULL, and LRGB datasets.
> > > >
> > > > 4. Table 1 is hard to read and understand at first glance. Please either highlight the best/second-best method or refrain from highlighting altogether.
> > > >
> > > > Regarding Table 1, we've mentioned in the caption that the highlighted entries indicate models that can count the substructure.
> > > >
> > > > 5. There are some language issues, which would require an additional pass over the paper (this is a minor point but it nevertheless slightly impacts accessibility).
> > > >
> > > > We have addressed the language issues that reviewers pointed out and will continue to work on further improvements in this regard.
> > > >
> > > >
> > > > Once again, thank you for your feedback, and we look forward to your continued evaluation of our work. Please feel free to reach out if you have any more questions or concerns.

---

> > > > > ### Comment · Reviewer_8zdN · 2023-11-23
> > > > >
> > > > > Thank you for your kind words, they are appreciated. At this point, I have no further questions or comments about the work.

---

### Official Review · Reviewer_knXN · 2023-10-31

**Soundness:** 3 good
**Presentation:** 3 good
**Contribution:** 3 good
**Rating:** 8
**Confidence:** 4

**Summary:**

The paper introduces a novel graph-to-set conversion that transforms interconnected nodes into a set of independent points, which allows seamless application of set encoder architectures such as the Transformer towards graph representation learning. The key is to perform symmetric rank decomposition on the adjacency or related matrices and process the coordinates through an orthogonal-transformation-equivariant set encoder, for which the paper proposes a new architecture, Point Set Transformer (PST). This approach has the benefit of not requiring any positional encodings used in previous graph Transformer literature, and is also theoretically shown to have stronger short-range and long-range expressivity compared to existing baselines. Experiments on substructure counting, molecular property prediction, and the Long Range Graph Benchmark validate the effectiveness of PST on various domains.

**Strengths:**

- [S1] The perspective of viewing graphs as point sets and performing graph representation learning via a set encoder is very interesting and original, and has great potential across the graph learning community.
- [S2] The idea is fairly simple and easy-to-follow, yet empirically effective as shown in the presented experimental results.

**Weaknesses:**

- [W1] There are a few details missing in the methodology/experiments that may help to clarify towards better reproducibility.
  - For each experiment, how is the rank $r$ chosen? Is it chosen via hyperparameter tuning? For larger graphs, it seems choosing a small $r$ will result in loss of information on the connectivity of the input graph. How is it that PST still performs well on Long Range Graph Benchmark despite this potential loss of information?
  - For the experiments, which matrix was used to generate the generalized coordinates? The adjacency matrix, or normalized adjacency, or some other matrix? The beginning of Section 4 briefly seems to mention that the method mainly uses the adjacency matrix, but this is not clear from the experimental section. Furthermore, an ablation study on which adjacency matrix works well with PST could be another interesting direction that provides further guidance.
  - After the graph-to-set conversion, how are edge features such as bond types in molecular graphs incorporated into PST?
- [W2] Another small concern is the computational cost due to use of symmetric rank decomposition. The paper mentions that each layer in PST runs in $\mathcal{O}(n^2 r)$-time and $\mathcal{O}(n^2 + nr)$-space, which can be costly if $n$ is large and $r$ must increase proportionally to $n$ in order to cover sufficient connectivity information. Table 10 in the Appendix shows runtime/memory consumption on ZINC, but since ZINC is consisted of fairly small graphs (~23.2 nodes per graph), it is unclear whether PST is scalable to large graphs (e.g. Long Range Graph Benchmark) as well. Could the authors elaborate on this?

**Questions:**

- Does the proposed graph-to-set conversion have any implications on graph generation [A, B, C] as well? Considering that the mapping is a bijection, being able to generate graphs via set generation would be another interesting direction, and any comments could further support the significance of the paper.
- Typo in end of Subsection 7.1: "(Theorem 6 and Theorem 6)" -> "(Theorem 6 and Theorem 7)"

[A] Kong et al., Autoregressive Diffusion Model for Graph Generation. (ICML 2023)\
[B] Vignac et al., Digress: Discrete Denoising Diffusion for Graph Generation. (ICLR 2023)\
[C] Jo et al., Score-based Generative Modeling of Graphs via the System of Stochastic Differential Equations. (ICML 2022)

---

> ### Author Response · Authors · 2023-11-17
> **Response to Reviewer knXN (1/2)**
>
> We are thankful for Reviewer knXN's comprehensive feedback and address their concerns as follows:
>
> - **W1-1:** For each experiment, how is the rank $r$ chosen? Is it chosen via hyperparameter tuning? For larger graphs, it seems choosing a small $r$ will result in loss of information on the connectivity of the input graph. How is it that PST still performs well on Long Range Graph Benchmark despite this potential loss of information?
>
>
> In our approach, $r$ is not a hyperparameter but the intrinsic rank of the decomposed matrix, depending solely on the data. This means there is no information loss associated with the choice of $r$.
>
> - **W1-2:** For the experiments, which matrix was used to generate the generalized coordinates? The adjacency matrix, or normalized adjacency, or some other matrix? The beginning of Section 4 briefly seems to mention that the method mainly uses the adjacency matrix, but this is not clear from the experimental section. Furthermore, an ablation study on which adjacency matrix works well with PST could be another interesting direction that provides further guidance.
>
> We clarify that various matrices, including the adjacency matrix, normalized adjacency, and Laplacian matrix, are theoretically viable for generating generalized coordinates. In our experiments, we primarily use the Laplacian matrix. This will be made clearer in our revised manuscript.
>
> We have added an ablation study on the choice of matrix for decomposition in the QM9 dataset. The results, displayed in the table below, compare PST variants using different matrices for decomposition against DF, our strongest baseline on the QM9 dataset. The similar performance across PST-Laplacian, PST-Adjacency, and PST-Normalized Adjacency illustrates that our method's effectiveness is not heavily dependent on the specific matrix chosen for decomposition.
>
> |  | $\mu$ | $\alpha$ | $\varepsilon_{\text{homo}}$ | $\varepsilon_{\text{lumo}}$ | $\Delta\varepsilon$ | $R^2$ | ZPVE | $U_0$ | $U$ | $H$ | $G$ | $C_v$ |
> | --- | --- | --- | --- | --- | --- | --- | --- | --- | --- | --- | --- | --- |
> | Unit| $10^{-1}$D | $10^{-1}$$a_0^3$ | $10^{-2}$meV | $10^{-2}$meV | $10^{-2}$meV | $a_0^2$ | $10^{-2}$meV | meV | meV | meV | meV | $10^{-2}$cal/mol/K |
> | PST-Laplacian | $3.19_{\pm 0.04}$ | $1.89_{\pm 0.04}$ | $5.98_{\pm 0.09}$ | $5.84_{\pm 0.08}$ | $8.46_{\pm 0.07}$ | $13.08_{\pm 0.16}$ | $0.39_{\pm 0.01}$ | $3.46_{\pm 0.17}$ | $3.55_{\pm 0.10}$ | $3.49_{\pm 0.20}$ | $3.55_{\pm 0.17}$ | $7.77_{\pm 0.15}$ |
> | PST-Adjacency | $3.16_{\pm 0.02}$ | $1.86_{\pm 0.01}$ | $6.31_{\pm 0.06}$ | $6.10_{\pm 0.05}$ | $8.84_{\pm 0.01}$ | $13.60_{\pm 0.09}$ | $0.39_{\pm 0.01}$ | $3.59_{\pm 0.12}$ | $3.73_{\pm 0.08}$ | $3.65_{\pm 0.06}$ | $3.60_{\pm 0.016}$ | $7.62_{\pm 0.21}$ |
> | PST-Normalized Adjacency | $3.22_{\pm 0.04}$ | $1.85_{\pm 0.02}$ | $5.97_{\pm 0.23}$ | $6.15_{\pm 0.07}$ | $8.79_{\pm 0.04}$ | $13.42_{\pm 0.15}$ | $0.41_{\pm 0.01}$ | $3.36_{\pm 0.25}$ | $3.41_{\pm 0.24}$ | $3.46_{\pm 0.18}$ | $3.38_{\pm 0.23}$ | $8.10_{\pm 0.12}$ |
> | DF(our strongest baseline on QM9) | $3.46$ | $2.22$ | $6.15$ | $6.12$ | $8.82$ | $15.04$ | $0.46$ | $4.24$ | $4.16$ | $3.95$ | $4.24$| $9.01$|

---

> ### Author Response · Authors · 2023-11-17
> **Response to Reviewer knXN (2/2)**
>
> - **W1-3** After the graph-to-set conversion, how are edge features such as bond types in molecular graphs incorporated into PST?
>
> We will make it more clear in Appendix F in revision. On the ZINC dataset, where edge features are scalar (representing bond types), these are directly considered in the adjacency and Laplacian matrices. For datasets with multi-dimensional edge features, where the adjacency matrix is represented as $A\in \mathbb{R}^{n\times n\times d}$, a theoretical approach is to decompose each channel separately, resulting in coordinates with $d$ channels $Q\in \mathbb{R}^{n\times m\times d}$. However, due to efficiency concerns, we first decompose the adjacency matrix without edge features to obtain coordinates $Q\in \mathbb{R}^{n\times m}$, then compute $A^{(i)}Q$ for each channel $i$ to generate the corresponding channel of the coordinates.
>
> - **W2** Another small concern is the computational cost due to use of symmetric rank decomposition. The paper mentions that each layer in PST runs in $O(n^2r)$-time and $O(n^2+nr)$-space, which can be costly if $n$ is large and $r$ must increase proportionally to $n$ in order to cover sufficient connectivity information. Table 10 in the Appendix shows runtime/memory consumption on ZINC, but since ZINC is consisted of fairly small graphs (~23.2 nodes per graph), it is unclear whether PST is scalable to large graphs (e.g. Long Range Graph Benchmark) as well. Could the authors elaborate on this?
>
> The scalability of our approach on large graphs, such as those in the Long Range Graph Benchmark (LRGB), is indeed a challenge. For example, on the Pascal-VOC SP dataset with ~400 nodes per graph, we limited the batch size to 6 and compare it with other GNNs. The results are as follows. We will include these results in Appendix I of our revised manuscript. To address scalability, we are exploring methods like linear transformers or sparse transformers for future enhancements of PST.
>
>
> |                           | GCN  | GPS   | SAN  | PST   |
> | ------------------------- | ---- | ----- | ---- | ----- |
> | training time per epoch/s | 77.2 | 246.2 | 270.2| 680.5 |
> | GPU memory/G              | 1.97 | 2.15  | 10.13| 18.95 |
>
> - **Q1:** Does the proposed graph-to-set conversion have any implications on graph generation [A, B, C] as well? Considering that the mapping is a bijection, being able to generate graphs via set generation would be another interesting direction, and any comments could further support the significance of the paper.
>
> Indeed, our graph-to-set conversion offers exciting prospects for graph generation. An orthogonal-transformation-equivariant diffusion model could generate node coordinates, from which the graph structure can be recovered. This approach aligns with techniques in molecule generation, where graphs represent atomic interconnections in 3D space. For instance, [1] utilizes an O(3)-invariant diffusion model for generating atomic coordinates. Our method can extend this idea to a broader range of graphs beyond molecules.
>
> [1] Emiel Hoogeboom et al. Equivariant Diffusion for Molecule Generation in 3D. ICML 2022.
>
> 6. **Q2** Typo in end of Subsection 7.1: "(Theorem 6 and Theorem 6)" -> "(Theorem 6 and Theorem 7)"
>
> We appreciate this catch and have corrected the typo in the revised manuscript.

---

> ### Comment · Reviewer_knXN · 2023-11-20
> **Response to Author Rebuttal by Reviewer knXN**
>
> Thank you authors for your commitment into sharing the rebuttal with additional results and clarifications.
>
> As shown in the runtime/memory measurements, scaling PST to large networks certainly seems to be a challenge, yet I agree with the authors that this can be addressed in the near future as there exist many approaches designed for making Transformers more efficient.
>
> After reading other reviewers' comments (as well as authors' respective responses), I still think incorporating SRD and an orthogonal transformation-invariant architecture such as PST for graph encoding is a neat approach towards maintaining symmetry under differences in eigenspaces, and the empirical results clearly show its effectiveness.
>
> All of my concerns have been addressed, and thus I retain my score as accept.

---

> > ### Author Response · Authors · 2023-11-21
> > **Response to Reviewer knXN**
> >
> > We thank the reviewer for thoroughly reading our responses and giving the affirmative feedbacks.

---

### Official Review · Reviewer_WPLr · 2023-11-01

**Soundness:** 2 fair
**Presentation:** 3 good
**Contribution:** 2 fair
**Rating:** 5
**Confidence:** 4

**Summary:**

The paper introduces a graph representation learning approach by converting interconnected graphs into sets of independent points and encoding them using an orthogonal-transformation-equivariant Transformer. This graph-to-set conversion method, based on Symmetric Rank Decomposition (SRD), eliminates the need for intricate positional encodings used in traditional graph Transformers. The authors theoretically demonstrate that two graphs are isomorphic if and only if their converted point sets are equal up to an orthogonal transformation. They also propose a parameterization of SRD using permutation-equivariant functions for practical implementation. The paper introduces a Point Set Transformer (PST) for encoding the transformed point set.

**Strengths:**

1. The authors provide theoretical foundations for their approach, demonstrating that isomorphic graphs can be perfectly represented using their method. The use of Symmetric Rank Decomposition is well-explained and supported by theorems. They also provide expressivity results regarding the short-range and the long-range abilities of the models.

2. The proposed method seems to outperform recent graph transformer architectures in the datasets used in this study.

**Weaknesses:**

1. One notable weakness in the paper is the tendency to overstate the novelty of certain ideas without providing adequate justification. Specifically, the paper emphasizes the concept of transforming a graph into a set of independent nodes as a novel approach. While the paper introduces a unique method of achieving this transformation through Symmetric Rank Decomposition (SRD), it does not sufficiently acknowledge that the idea of treating graphs as sets of nodes is not entirely novel in the context of graph neural networks. Many existing graph neural network models, including graph transformers, do employ the idea of viewing graphs as sets of nodes for processing. They use various techniques, including positional encodings, to capture and leverage the graph's structural information. While the paper rightfully introduces SRD as a different method for this transformation, it should clarify why this particular approach is advantageous or provides a significant improvement over existing methods. This would help readers understand the specific contributions and benefits of the proposed approach more clearly. Furthermore, the paper mentions the elimination of positional encodings in the proposed method, implying that this is beneficial. However, it does not provide a comprehensive explanation or empirical evidence to support this claim. The paper should clarify why removing positional encodings is advantageous and how this contributes to the overall effectiveness of the proposed approach. Without a clear rationale or evidence, it leaves readers questioning the choice to eliminate positional encodings and the potential impact on performance. In summary, the paper should provide a more balanced perspective on the novelty of its ideas and offer a robust justification for choices such as removing positional encodings to enhance the clarity and credibility of its contributions.

2. The paper does not present empirical results from a wide range of graph benchmarks, which could have provided a more comprehensive assessment of the method's performance and generalizability. TUDatasets, like many other benchmark datasets, cover diverse graph structures and characteristics, and their inclusion in the evaluation process would have added valuable insights into the method's effectiveness across different graph types.

**Questions:**

1. It would be beneficial to see experimental results on a variety of benchmark datasets, including TUDatasets, to assess the method's performance and how it compares to existing approaches. Can the authors provide results on more graph benchmarks to substantiate their claims?

2. The paper suggests that removing positional encodings is beneficial, but the rationale behind this choice is not clearly explained. Can the authors elaborate on why the elimination of positional encodings is advantageous and how it contributes to the overall performance of the method?

---

> ### Author Response · Authors · 2023-11-17
> **Response to Reviewer WPLr (1/2)**
>
> We are grateful for Reviewer WPLr's comprehensive feedback and address their concerns as follows:
>
> - **W1:** One notable weakness in the paper is the tendency to overstate the novelty of certain ideas without providing adequate justification.   While the paper introduces a unique method of achieving this transformation through Symmetric Rank Decomposition (SRD), it does not sufficiently acknowledge that the idea of treating graphs as sets of nodes is not entirely novel in the context of graph neural networks. Many existing graph neural network models, including graph transformers, do employ the idea of viewing graphs as sets of nodes for processing. They use various techniques, including positional encodings, to capture and leverage the graph's structural information. While the paper rightfully introduces SRD as a different method for this transformation, it should clarify why this particular approach is advantageous or provides a significant improvement over existing methods. This would help readers understand the specific contributions and benefits of the proposed approach more clearly.
>
> Though we agree that existing graph transformers and ordinary GNNs also encode the nodes, we respectfully argue that our assertion of a "paradigm shift" is not an overstatement. Our innovation is not merely SRD but an entirely different architecture. Existing models still perceive nodes as **interconnected**, relying on adjacency matrices or their variants, such as random walk matrices, for input. Our approach represents a significant divergence from this norm. By converting graphs to sets of **independent** nodes and utilizing a set encoder without adjacency inputs, we shift the paradigm from encoding adjacency matrices to crafting set encoders for independent nodes.  This paradigm shift, which completely gives up adjacency input, enables **a novel model design space** too. For example, message passing or attention matrix with adjacency information as relative positional encoding is no longer needed---a set encoder over independent points is enough.
>
> The paradigm shift is rooted in a fundamental difference between SRD and eigendecomposition (EVD), which is commonly used in prior works. As explicated in our Proposition 1, the symmetric rank decomposition (SRD) of a matrix is uniquely determined up to an orthogonal transformation of the entire coordinate system. In contrast, as elucidated in Section 2 of [1], EVD is uniquely determined up to orthogonal transformations within each eigenspace. This fundamental difference allows SRD-based models to effortlessly maintain symmetry (yielding consistent predictions for isomorphic graphs), while EVD-based methods must address each eigenspace individually, making existing EVD-based methods such as [1] hardly work on graph-level tasks where eigenspaces differ between graphs. In essence, SRD leads to a practical **bijective** conversion of graph problems into set problems where a unified set learner can be applied to different graphs, a property which is not achievable by existing EVD methods.
>
> Furthermore, our method distinctively maintains invariance, producing identical predictions for isomorphic graphs, a feature not consistently achieved by most existing positional encoding methods, such as direct utilization of eigenvectors [2]. This attribute enhances the generalization capabilities of our model.

---

> > ### Author Response · Authors · 2023-11-17
> > **Response to Reviewer WPLr (2/2)**
> >
> > - **W1-2 & Q2:** Furthermore, the paper mentions the elimination of positional encodings in the proposed method, implying that this is beneficial. However, it does not provide a comprehensive explanation or empirical evidence to support this claim. The paper should clarify why removing positional encodings is advantageous and how this contributes to the overall effectiveness of the proposed approach. Without a clear rationale or evidence, it leaves readers questioning the choice to eliminate positional encodings and the potential impact on performance. // The paper suggests that removing positional encodings is beneficial, but the rationale behind this choice is not clearly explained. Can the authors elaborate on why the elimination of positional encodings is advantageous and how it contributes to the overall performance of the method?
> >
> > It seems there has been a misunderstanding regarding our stance on positional encodings. We recognize the effectiveness of positional encodings in existing GNNs, as evidenced in [1]. Our model, being a transformer, can indeed incorporate positional encodings, potentially leading to performance gains. However, our primary aim is to introduce a new paradigm for Graph Learning, rather than merely presenting a new state-of-the-art graph transformer with various positional encodings. By omitting positional encodings, we highlight the inherent advantages of our "graph as set" approach. In our experiments, we demonstrate that our PST without positional encodings outperforms graph transformers that include various positional encodings, thereby validating our approach.
> >
> >
> > - **Q3:** The paper does not present empirical results from a wide range of graph benchmarks, which could have provided a more comprehensive assessment of the method's performance and generalizability. TU Datasets, like many other benchmark datasets, cover diverse graph structures and characteristics, and their inclusion in the evaluation process would have added valuable insights into the method's effectiveness across different graph types.
> >
> > We appreciate this suggestion and have expanded our empirical evaluation to include a broader range of graph benchmarks. Our initial experiments have covered a diverse set of datasets, including substructure counting, ZINC, ZINC-full, ogbg-molhiv, qm9, and long-range graph benchmarks, all of which are widely recognized in GNN research. In response to the reviewer's feedback, we have now included additional experiments on four TU datasets, detailed in Appendix J of our revised manuscript. The results, as presented in the table below, demonstrate that our PST consistently outperforms baseline models across these datasets, highlighting its effectiveness and generalizability.
> >
> > | Method       | MUTAG        | PTC-MR       | PROTEINS     | IMDB-B       |
> > | ------------ | ------------ | ------------ | ------------ | ------------ |
> > | WL           | 90.4±5.7     | 59.9±4.3     | 75.0±3.1     | 73.8±3.9     |
> > | GIN          | 89.4±5.6     | 64.6±7.0     | 75.9±2.8     | 75.1±5.1     |
> > | DGCNN        | 85.8±1.7     | 58.6 ±2.5    | 75.5±0.9     | 70.0±0.9     |
> > | GraphSNN     | 91.2±2.5     | 67.0±3.5     | 76.5±2.5     | 76.9±3.3     |
> > | GIN-AK+      | 91.3±7.0     | 68.2±5.6     | 77.1±5.7     | 75.6±3.7     |
> > | KP-GCN       | 91.7±6.0     | 67.1±6.3     | 75.8±3.5     | 75.9±3.8     |
> > | KP-GraphSAGE | 91.7±6.5     | 66.5±4.0     | 76.5±4.6     | 76.4±2.7     |
> > | KP-GIN       | 92.2±6.5     | 66.8±6.8     | 75.8±4.6     | 76.6±4.2     |
> > | PST          | **94.4±3.5** | **68.8±4.6** | **80.7±3.5** | **78.9±3.6** |
> >
> > These results reinforce the strengths of our approach and its applicability across a diverse array of graph structures and characteristics.
> >
> > [1] Sign and Basis Invariant Networks for Spectral Graph Representation Learning, Lim et al., ICLR 2023.
> >
> > [2] Benchmarking Graph Neural Networks, JMLR 2023.

---

> > > ### Author Response · Authors · 2023-11-22
> > > **Urgent Response Requested: The last day of discussion stage.**
> > >
> > > Dear Reviewer WPLr,
> > >
> > > We would like to express our gratitude for your valuable feedback and constructive comments on our paper. Your insights have significantly contributed to improving the quality and clarity of our work.
> > >
> > > We have carefully considered and addressed all of your comments and suggestions in our response. However, we have not received any further communication from you since our last response. Given the importance of your feedback to the finalization of our paper, we kindly request your prompt response.
> > >
> > > To summarize our responses:
> > >
> > > W1: We clarified our novelty and distinction from existing works. Our clarification "paradigm shift" is not an overstatement.
> > >
> > > Q3: We have included experiments on TU datasets. Our model outperforms existing models.
> > >
> > > Thank you once again for your time and effort in reviewing our work. We look forward to hearing from you soon.
> > >
> > > Best, Authors

---

> > > > ### Author Response · Authors · 2023-11-23
> > > > **Looking Forward to Your Reply**
> > > >
> > > > Dear Reviewer WPLr,
> > > >
> > > > We genuinely appreciate your meticulous review of our paper. We have taken great care to thoroughly address all of your valuable comments and suggestions in our response. However, we have yet to receive any additional communication from you, and with only 3 hours remaining for the discussion phase, your input is crucial to ensure a fair evaluation of our work. We eagerly await your response and hope to hear from you soon.
> > > >
> > > > Best, Authors`

---

### Official Review · Reviewer_h4E8 · 2023-11-06

**Soundness:** 3 good
**Presentation:** 3 good
**Contribution:** 2 fair
**Rating:** 5
**Confidence:** 3

**Summary:**

This paper proposes a new architecture for graph representation learning. Specifically, the proposed model uses symmetric rank decomposition to obtain coordinates for each node in the graph. Then a transformer model treats the graph as a set of nodes (augmented with coordinates) and encodes the set. Since the coordinates are up to transformations by orthogonal matrices, the transformer model is designed to be invariant to orthogonal transformations. This paper analyses the expressive power of the model, and the model shows good empirical performance compared with existing graph transformers.

**Strengths:**

- A clear presentation of the methodology; easy to follow
- The design of the architecture is well motivated by the analysis of symmetric rank decomposition, and this paper further provides theoretical analysis of expressiveness
- The proposed model shows good empirical performance

**Weaknesses:**

- This paper exaggerates its contributions: this paper claims that the proposed model is a "paradigm shift", but the use of symmetric rank decomposition is actually not much different from existing positional encodings based on eigendecomposition. Also, the invariance to orthogonal transformations of positional encoding is discussed and addressed by several related papers (e.g., [1]). I don't think the proposed model is significantly different.
- Given the above point, the experiment part should also include the performance of [1] and do a comparison, but it's missing.
- Another recent graph transformer is missing [2] which seems to have better performance.

[1] Sign and Basis Invariant Networks for Spectral Graph Representation Learning. Lim et al. ICLR 2023

[2] Graph Inductive Biases in Transformers without Message Passing. ICML 2023

**Questions:**

Please see Weakness

---

> ### Author Response · Authors · 2023-11-17
> **Response to Reviewer h4E8 (1/2)**
>
> We appreciate the detailed feedback from reviewer h4E8 and address their concerns as follows:
>
> - **W1-1:** This paper exaggerates its contributions: this paper claims that the proposed model is a "paradigm shift", but the use of symmetric rank decomposition is actually not much different from existing positional encodings based on eigendecomposition.
>
> Thank you for the valuable comment. We respectfully argue that our assertion of a "paradigm shift" is not an overstatement. Below are detailed reasons.
>
> **Distinct Properties of SRD and EVD:** As explicated in our Proposition 1, the symmetric rank decomposition (SRD) of a matrix is uniquely determined up to an orthogonal transformation of the entire coordinate system. In contrast, as elucidated in Section 2 of [1], eigendecomposition (EVD) is uniquely determined up to orthogonal transformations within each eigenspace. This fundamental difference allows SRD-based models to effortlessly maintain symmetry (yielding consistent predictions for isomorphic graphs), while EVD-based methods must address each eigenspace individually, making existing EVD-based methods such as [1] hardly work on graph-level tasks where eigenspaces differ between graphs. In essence, SRD leads to a practical **bijective** conversion of graph problems into set problems where a unified set learner can be applied to different graphs, a property which is not achievable by existing EVD methods.
>
> **Paradigm Shift:** Owing to this property difference, our model represents a significant departure from existing methods. Contrary to positional encoding methods that require **GNNs with inter-node edges** as input (even Graph Transformers still use adjacency matrices or their variants), our approach converts the graph into a **set of independent nodes** and employs a set encoder **without adjacency inputs**. This shift from encoding adjacency matrices of interconnected nodes to using set encoders for independent nodes, devoid of edge information, signifies a major deviation from traditional methodologies. This paradigm shift, which completely gives up adjacency input, enables **a novel model design space** too. For example, message passing or attention matrix with adjacency information as relative positional encoding is no longer needed---a set encoder over independent points is enough.
>
> **Maintaining Invariance:** Our method distinctively maintains invariance, producing identical predictions for isomorphic graphs, a feature not consistently achieved by most existing positional encoding methods, such as direct utilization of eigenvectors [3]. This attribute enhances the generalization capabilities of our model.
>
> - **W1-2:**  Also, the invariance to orthogonal transformations of positional encoding is discussed and addressed by several related papers (e.g., [1]). I don't think the proposed model is significantly different.
>
> We agree that both our work and [1] address the concept of invariance to orthogonal transformations. Nevertheless, there are significant differences between the two approaches, encompassing the fundamental properties of the two decompositions, the resulting invariance, the architectural designs, and the overarching paradigms.
>
> **Nature of Orthogonal Transformations:** Although both works refer to orthogonal transformations, the contexts in which they are applied differ markedly. In our approach using symmetric rank decomposition (SRD), the orthogonal transformation is applied to the entire coordinate system. In contrast, the orthogonal transformation in the eigendecomposition (EVD) used in [1] is specific to each eigenspace.
>
> **Addressing Orthogonal Invariance:** This fundamental distinction in decomposition means that [1] **cannot completely solve the issue of orthogonal invariance.** Our model can directly integrate an invariant set encoder with the coordinates generated by our SRD. Conversely, [1] is compelled to address each eigenspace individually, resulting in an architecture (BasisNet) that is only applicable to graphs with a fixed number of multiple eigenvalues. This limitation is impractical for graph tasks with varying numbers of multiple eigenvalues. For real-world datasets, [1] is forced to compromise on orthogonal invariance by assuming a uniform multiplicity of 1 for all eigenvalues, as seen in SignNet.
>
> **Divergent Network Structures:** The above difference also leads to distinct network structures. SignNet and BasisNet in [1] require specialized blocks like invariant graph networks and address eigenvectors in each eigenspace separately. In contrast, our Point Set Transformer (PST) more closely resembles a conventional transformer. In fact, our paradigm only requires a set encoder invariant to orthogonal transformations, thus enabling very flexible model choices not limited to the transformer which we implement in this paper.

---

> ### Author Response · Authors · 2023-11-17
> **Response to Reviewer h4E8 (2/2)**
>
> - **W2 & W3:** Given the above point, the experiment part should also include the performance of [1] and do a comparison, but it's missing. Another recent graph transformer is missing [2] which seems to have better performance.
>
> We thank the reviewer for these constructive suggestions. In response, we have updated our manuscript to include these comparisons.
>
> **Comparison with [1]:** Our PST demonstrates a substantial improvement over [1], with a notable 30% margin in performance on both the ZINC and ZINC-full datasets.
>
> **Comparison with [2]:** Our PST shows competitive performance relative to [2]. On the ZINC dataset, [2] marginally surpasses our model by 9%. However, our PST excels on the larger ZINC-full dataset, outperforming [2] by 33%. Both models achieve nearly identical results on the LRGB datasets.
>
>
> References:
>
> [1] Sign and Basis Invariant Networks for Spectral Graph Representation Learning, Lim et al., ICLR 2023.
>
> [2] Graph Inductive Biases in Transformers without Message Passing, ICML 2023.
>
> [3] Benchmarking Graph Neural Networks, JMLR 2023.

---

> > ### Author Response · Authors · 2023-11-22
> > **Urgent Response Requested: The last day of discussion stage.**
> >
> > Dear Reviewer h4E8,
> >
> > We would like to express our gratitude for your valuable feedback and constructive comments on our paper. Your insights have significantly contributed to improving the quality and clarity of our work.
> >
> > We have carefully considered and addressed all of your comments and suggestions in our response. However, we have not received any further communication from you since our last response. Given the importance of your feedback to the finalization of our paper, we kindly request your prompt response.
> >
> > To summarize our responses:
> >
> > W1: We clarified our novelty and distinction from existing works. Our clarification "paradigm shift" is not an overstatement.
> >
> > W2&W3: We have included the baselines. Our model performs comparable or better than them.
> >
> > Thank you once again for your time and effort in reviewing our work. We look forward to hearing from you soon.
> >
> > Best, Authors

---

> > > ### Author Response · Authors · 2023-11-23
> > > **Looking Forward to Your Reply**
> > >
> > > Dear Reviewer h4E8,
> > >
> > > We genuinely appreciate your meticulous review of our paper. We have taken great care to thoroughly address all of your valuable comments and suggestions in our response. However, we have yet to receive any additional communication from you, and with only 3 hours remaining for the discussion phase, your input is crucial to ensure a fair evaluation of our work. We eagerly await your response and hope to hear from you soon.
> > >
> > > Best, Authors`

---

> > > > ### Comment · Reviewer_h4E8 · 2023-11-23
> > > > **Comment by Reviewer**
> > > >
> > > > Thank you for the further clarification! I understand the difference between PST and Sign/BasisNet, and I would say PST is a reasonable improvement over Sign/BasisNet. However, I still don't agree with the "Paradigm Shift" claim.
> > > >
> > > > Actually, existing graph transformers don't need adjacency input either. The positional encodings are pre-computed and then added/appended to node features, and then nodes are encoded as a set. PST proposed in this paper needs to pre-compute SRD from the adjacency matrix, and the output of SRD is actually an instantiation of positional encoding.
> > > >
> > > > Considering the above connection to existing works, I think this paper overstates its contribution. A similar concern is also raised by Reviewer WPLr. Therefore I will keep my rating.

---

> ### Author Response · Authors · 2023-11-23
> **Graph transformers need adjacency input**
>
> Thank you for your thoughtful feedback and for considering our response. We appreciate your acknowledgment of the improvements our proposed methodology brings over existing approaches. However, we would like to further clarify why we believe our work constitutes a "paradigm shift" in the context of graph transformers.
>
> * First, we would like to point out that **While it is true that existing graph transformers utilize positional encodings to capture graph structure, they typically still rely on the adjacency input.** Specifically,
>
> 1. Grit [1] and graphit [8] uses random walk matrix (normalized adjacency matrix) as relative positional encoding.
> 2. Vanilla Graph Transformer [6], Graphormer [2] and SAN [3] uses adjacency matrix as relative positional encoding. [2]'s ablation shows that adjacency is crucial.
> 3. GPS [4], Exphormer [5], higher-order transformer [7], Graph Vit/MLP-Mixer [9] even directly incorporating message passing block which use adjacency matrix to guide message passing between node.
>
> Therefore, our method without adjacency input differs from existing models in architecture significantly.
>
> * We emphasize that this architecture distinction is not just a minor improvement but a substantial departure from the prevailing approach. By doing so, we enable **a different design space for GNN, where set encoders can be applied to graph tasks easily**, opening up opportunities for novel applications and addressing potential limitations associated with adjacency-based methods. For example, we can also employ set encoders other than transformer like deepset, as demonstrated in our response to Reviewer 8dzn's
>
> In summary, we believe that our work, using set encoder without adjacency input for graph task, represents a **significant paradigm shift**, and we hope that you will reconsider our claim in light of this explanation. We value your perspective and look forward to any further discussion or feedback you may have.
>
> Best, Authors.
>
> [1] Liheng Ma et al. Graph Inductive Biases in Transformers without Message Passing. ICML 2023.
>
> [2] Chengxuan Ying et al. Do Transformers Really Perform Badly for Graph Representation? NeurIPS 2021.
>
> [3] Devin Kreuzer et al. Rethinking Graph Transformers with Spectral Attention. NeuIPS 2021.
>
> [4] Ladislav Rampásek et al. Recipe for a General, Powerful, Scalable Graph Transformer. NeurIPS 2022
>
> [5] Hamed Shirzad et al. Exphormer: Sparse Transformers for Graphs. ICML 2023.
>
> [6] Vijay Prakash Dwivedi, Xavier Bresson. A Generalization of Transformer Networks to Graphs. https://arxiv.org/abs/2012.09699.
>
> [7] Jinwoo Kim et al. Transformers Generalize DeepSets and Can be Extended to Graphs and Hypergraphs. NeurIPS, 2021.
>
> [8] Gregoire Mialon et al, GraphiT: Encoding Graph Structure in Transformers. arXiv/2106.05667.
>
> [9] Xiaoxin He et al. A Generalization of ViT/MLP-Mixer to Graphs. ICML 2023.

---

### Author Response · Authors · 2023-11-20
**Looking Forward to Your Reply**

Dear Reviewers,

Thanks for your time in reviewing our paper. We got many constructive questions and valuable feedback, and have answered these questions in detail. However, we have not received your responses. The discussion stage only has three days left. Could you please take some time to read our rebuttal?

We are looking forward to your further comment on our work.

Best, Authors

---

> ### Author Response · Authors · 2023-11-21
> **Discussion Stage Closing Soon**
>
> Dear Reviewers,
>
> We sincerely appreciate your dedicated time and effort in reviewing our paper. As the discussion stage is rapidly coming to a close, we kindly request your prompt attention to our rebuttal. Your valuable insights and feedback are integral to ensuring a fair and comprehensive evaluation of our paper.
>
> We eagerly anticipate your continued input on our work. If you still have any lingering questions or unresolved concerns, please don't hesitate to reach out. We are more than willing to provide further clarification and address any issues that may arise.
>
> Thank you once again for your invaluable contributions to the review process.
>
> Many thanks, Authors

---

### Meta-Review · Area_Chair_BB88 · 2023-12-09

**Metareview:**

This paper proposes to study graphs as point sets. While the reviewers acknowledge some positive aspects of this work, they are concerned with the overclaim of this work in terms of novelty compared to existing literature. Thus a reject is recommended so that the authors can revise their paper accordingly.

**Justification For Why Not Higher Score:**

The reviewers are concerned with the overclaim of this work in terms of novelty compared to existing literature.

**Justification For Why Not Lower Score:**

N/A

---

### Decision · Program_Chairs · 2024-01-16

Reject